# Kaleido Diffusion: Improving Conditional Diffusion Models with Autoregressive Latent Modeling

**Jiatao Gu**[†*]**, Ying Shen**[◇*]**, Shuangfei Zhai**[†]**, Yizhe Zhang**[†]**, Navdeep Jaitly**[†]**, Josh Susskind**[†]

[†]Apple   [◇]University of Illinois Urbana-Champaign      [*] equal contribution

[†]{jgu32, szhai, yizzhang,njaitly, jsusskind}@apple.com  [◇]ying22@illinois.edu

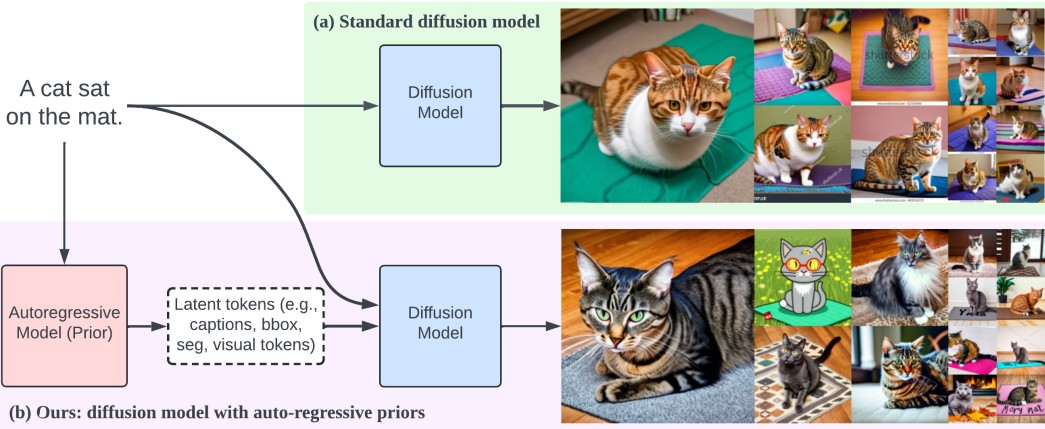

Figure 1: Comparison of the generated image samples given the caption "a cat sat on the mat". Our models generate more diverse images with the help of autoregressive latent modeling.

## Abstract

Diffusion models have emerged as a powerful tool for generating high-quality images from textual descriptions. Despite their successes, these models often exhibit limited diversity in the sampled images, particularly when sampling with a high classifier-free guidance weight. To address this issue, we present Kaleido, a novel approach that enhances the diversity of samples by incorporating autoregressive latent priors. Kaleido integrates an autoregressive language model that encodes the original caption and generates latent variables, serving as abstract and intermediary representations for guiding and facilitating the image generation process. In this paper, we explore a variety of discrete latent representations, including textual descriptions, detection bounding boxes, object blobs, and visual tokens. These representations diversify and enrich the input conditions to the diffusion models, enabling more diverse outputs. Our experimental results demonstrate that Kaleido effectively broadens the diversity of the generated image samples from a given textual description while maintaining high image quality. Furthermore, we show that Kaleido adheres closely to the guidance provided by the generated latents, demonstrating its capability to effectively control the image generation process.

## 1   Introduction

Diffusion models have become pervasive in many text-to-image generation tasks for their ability to generate high-quality images based on textual descriptions. A pivotal mechanism in these models is

38th Conference on Neural Information Processing Systems (NeurIPS 2024).

classifier-free guidance (CFG) [Ho and Salimans, 2021], which effectively steers the sampling process towards better alignment to textual prompts and improved sampling quality at the same time. CFG can be interpreted as tuning the temperature of the conditional distribution, whereas increasing the guidance scale sharpens the conditional distribution. This guides the generation to focus on regions of high conditional probability, effectively reducing sampling noise which is typically of lower density. However, while high CFG improves sampling quality, it simultaneously narrows the diversity in the generated samples. This manifests in the models' inability to produce diverse images from the same caption, even when there are variations in the initial noise that seeds the generation process. For instance, given a fixed textual description, "a cat sits on a mat", existing text-to-image diffusion models predominantly produce image samples depicting cats with similar colors and patterns, as illustrated in Figure 1. Such limited visual diversity hinders the practical application of diffusion models in scenarios where a wide range of creative and diverse visual interpretations are desired from identical textual inputs. It also poses challenges in scenarios demanding the representation of underrepresented data or accommodating a wide range of user preferences. Therefore, enhancing diversity in diffusion models without compromising the quality remains a critical research problem.

To tackle this, we introduce Kaleido, a general framework that improves diffusion models with autoregressive priors. Kaleido first defines a discrete encoding of images (eg, detailed captioning, bounding boxes), which captures desirable abstractions of images that's not included in the default text prompts. Next, Kaleido integrates an encoder-decoder language model that encodes the original text caption and autoregressively predicts the discrete latent tokens. Lastly, the diffusion model is conditioned on both the original text prompt and the autoregressively generated discrete latents and generates an image. This enriched conditioning allows Kaleido to produce a more diverse array of high-quality images, even at high guidance scales. We explore various forms of latents, including textual descriptions, detection bounding boxes, object blobs, and abstract visual tokens – all designed to refine and guide the conditional image generation process.

We experiment on both class and text conditioned image generation benchmarks [1]. We show that Kaleido not only outperforms standard diffusion models in terms of diversity but also maintains the high quality of the generated image. Additionally, the generated latents effectively control the characteristics of the generated images, ensuring that the image samples closely align with the intended latent variables. This modeling of latent tokens not only increases the diversity of image outputs but also provides a degree of interpretability and control over the image generation process.

To summarize, Kaleido exhibits the following advantages:

1. Kaleido promotes the diversity in generated image samples even with high CFG, allowing the image generation of both high quality and diversity.

2. The generated latent variables are interpretable, offering an explainable mechanism behind the image generation process, and facilitating an understanding of how different latents affect the outputs.

3. Kaleido provides a fine-grained, editable interface that allows users to adjust the discrete latent codes before final image production, granting greater flexibility and control over the output.

## 2 Preliminaries

**Autoregressive Image Generation**    The success of large language models (LLMs) in NLP has demonstrated their *scalability* and *universality* of modeling any complex data, motivating the development of using autoregressive models for image generation. Typically, autoregressive image generation operates on discrete image tokens obtained from vector-quantization (VQ) [Van Den Oord et al., 2017]. More precisely, given an image $x \in \mathbb{R}^{3 \times H \times W}$, we first obtain a sequence of discrete tokens $z_{1:N} = \mathcal{E}(x)$ which approximately reconstructs the input with a learned decoder $\mathcal{D}(z_{1:N}) \approx x$. Then, an autoregressive model is learned to predict the discrete tokens one after another, mirroring the sequential language modeling:

$$\mathcal{L}_\theta^{\text{AR}} = \sum_{n=1}^{N} \log P_\theta(z_n | z_{0:n-1}, c),  \tag{1}$$

where $c$ is the condition (e.g., class, text prompt, etc.), and $z_0$ is a special start token. At inference time, we first sample from the learned distribution, and then pass the sampled latents to the decoder

---

[1] the class conditioning setting can be considered as a special case of text conditioning.

($\mathcal{D}$) to get the final output. Such VQ-based paradigm has been the foundation for various text-to-image [Esser et al., 2021, Yu et al., 2021, Zheng et al., 2022, Yu et al., 2022] and multi-modal generation [Team et al., 2023, Team, 2024].

However, these methods share a common limitation: they primarily rely on discretization, which struggles to capture all the nuances of an image when using a limited length of discrete image token sequence. To generate higher-resolution images, a longer sequence of image tokens is necessary. Yet, this inherently leads to increased capacity demands. For instance, Yu et al. [2022] requires 20B parameters to work properly. Additionally, the left-to-right properties of these autoregressive models prevent the rewriting of previously generated image tokens, resulting in suboptimal image quality.

**Diffusion-based Image Generation**   Diffusion models [Sohl-Dickstein et al., 2015, Ho et al., 2020] are latent variable models with a pre-determined posterior distribution and are trained using a denoising objective, which has quickly become the new *de-facto* approach for image generation. Unlike autoregressive models which predict images as a sequence, diffusion-based models iteratively generate the whole image in a non-autoregressive fashion. Specifically, given an image $\boldsymbol{x} \in \mathbb{R}^{3 \times H \times W}$ and a signal-noise schedule $\{\alpha_t, \sigma_t\}$ where the signal-to-noise ratio (SNR) $(\alpha_t^2/\sigma_t^2)$ decreases monotonically with $t$, we define a series of latent variables $\boldsymbol{x}_t, t = 0, \ldots, T$ that adhere to:

$$q(\boldsymbol{x}_t|\boldsymbol{x}) = \mathcal{N}(\boldsymbol{x}_t; \alpha_t\boldsymbol{x}, \sigma_t^2 I), \text{ and } q(\boldsymbol{x}_t|\boldsymbol{x}_s) = \mathcal{N}(\boldsymbol{x}_t; \alpha_{t|s}\boldsymbol{x}_s, \sigma_{t|s}^2 I), \tag{2}$$

where $\boldsymbol{x}_0 = \boldsymbol{x}$, $\alpha_{t|s} = \alpha_t/\alpha_s$, and $\sigma_{t|s}^2 = \sigma_t^2 - \alpha_{t|s}^2\sigma_s^2$ for $s < t$. The model then learns to reverse this process using a backward model $p_\theta(\boldsymbol{x}_s|\boldsymbol{x}_t, \boldsymbol{c})$, which reformulates a denoising objective:

$$\mathcal{L}_\theta^{\text{DM}} = \mathbb{E}_{t \sim [1,T], \boldsymbol{x}_t \sim q(\boldsymbol{x}_t|\boldsymbol{x})} \left[\omega_t \cdot ||\boldsymbol{x}_\theta(\boldsymbol{x}_t, \boldsymbol{c}) - \boldsymbol{x}||_2^2\right], \tag{3}$$

where $\boldsymbol{x}_\theta(\boldsymbol{x}_t, \boldsymbol{c})$ is a neural network (typically a UNet [Ronneberger et al., 2015] or Transformer [Peebles and Xie, 2022]) that maps the noisy input $\boldsymbol{x}_t$ to its clean version $\boldsymbol{x}$, based on the time step $t$ and conditional input $\boldsymbol{c}$; $\omega_t \in \mathbb{R}^+$ is a loss weighting factor. In practice, $\boldsymbol{x}_\theta$ can be re-parameterized with noise- or v-prediction [Salimans and Ho, 2022] for enhanced performance, and can be applied on raw pixel space [Saharia et al., 2022, Gu et al., 2023] or latent space [Rombach et al., 2022].

**Classifier-free Guidance**   An intriguing property of conditional diffusion models is that we can easily guide the iterative sampling process for better sampling quality. For instance, Ho and Salimans [2021] introduced *Classifier-free Guidance (CFG)*, which utilizes the diffusion model itself to perform guidance at test time. More specifically, we perform sampling using the following linear combination:

$$\tilde{\boldsymbol{x}}_\theta(\boldsymbol{x}_t, \boldsymbol{c}) = \gamma \cdot (\boldsymbol{x}_\theta(\boldsymbol{x}_t, \boldsymbol{c}) - \boldsymbol{x}_\theta(\boldsymbol{x}_t)) + \boldsymbol{x}_\theta(\boldsymbol{x}_t), \tag{4}$$

where $\gamma$ is the guidance weight, and $\boldsymbol{x}_\theta(\boldsymbol{x}_t) = \boldsymbol{x}_\theta(\boldsymbol{x}_t, \boldsymbol{c} = \emptyset)$ is the unconditional denoising output. During training, we drop the condition $\boldsymbol{c}$ with certain probability $p_{\text{uncond}}$ to facilitate unconditional prediction. When $\gamma > 1$, CFG takes effect and amplifies the difference between conditional and unconditional generation, leading to a global control of high-quality generation.

Compared to autoregressive models, diffusion models are more flexible in adjusting sample steps, allowing for the utilization of noise schedules to learn different frequencies. Additionally, with the use of CFG, diffusion models can achieve higher quality images with much fewer parameters than autoregressive models. However, it's notable that CFG can significantly impact the diversity of the diffusion output, which motivates us to revisit the basics and combine the strengths of both.

# 3   Kaleido Diffusion

We propose Kaleido, a general framework that integrate an autoregressive prior with diffusion model to enhance image generation. As illustrated in Fig. 2, Kaleido comprises two major components: an AR model that generates latent tokens as abstract representations, and a latent-augmented diffusion model that iteratively synthesizes images based on these latents together with the original condition. For following sections, we first describe the importance to introduce additional latents in standard diffusion models (§ 3.1), and show how we can model them with AR models (§ 3.2). The training and inference procedure are described in § 3.3 and § 3.4.

## 3.1   Latent-augmented Diffusion Models

As demonstrated in Ho and Salimans [2021], diffusion with CFG (Eq. (4)) is equivalent to follow

$$\nabla_{\boldsymbol{x}} \log \tilde{p}_\theta(\boldsymbol{x}|\boldsymbol{c}) = \gamma \left[\nabla_{\boldsymbol{x}} (\log p_\theta(\boldsymbol{x}|\boldsymbol{c}) - \log p_\theta(\boldsymbol{x}))\right] + \nabla_{\boldsymbol{x}} \log p_\theta(\boldsymbol{x}), \tag{5}$$

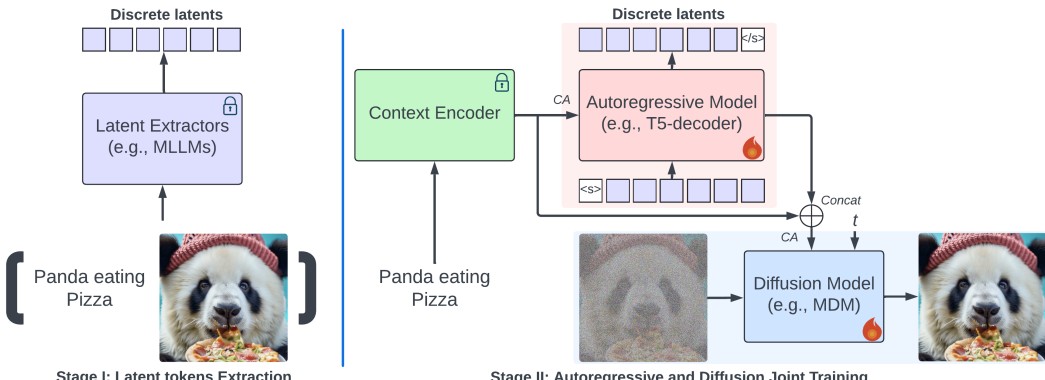

Figure 2: Training pipeline of the proposed Kaleido diffusion.

which can be interpreted as sampling from a "temperature-adjusted" distribution:

$$\boldsymbol{x} \sim \tilde{p}_\theta(\boldsymbol{x}|\boldsymbol{c}) \propto p_\theta(\boldsymbol{x})\left[p_\theta(\boldsymbol{c}|\boldsymbol{x})\right]^\gamma, \quad \text{where} \quad p_\theta(\boldsymbol{c}|\boldsymbol{x}) \propto p_\theta(\boldsymbol{x}|\boldsymbol{c})/p_\theta(\boldsymbol{x}). \tag{6}$$

Here $\gamma$ can be seen as inverse temperature, which sharpens the conditional distribution $p_\theta(\boldsymbol{c}|\boldsymbol{x})$ when $\gamma > 1$. That is to say, CFG is crucial as it guides the generation to only focus on high-probability regions, avoiding sampling noise (which tends to have low density). However, sharpening the distribution also reduces the diversity, causing undesirable phenomena like "mode collapse". This is because $\boldsymbol{c}$ (e.g., class label, text prompt, etc.) normally does not contain all the information that describes $\boldsymbol{x}$. Suppose we introduce a hypothetical variable $\boldsymbol{z}$ to represent the "modes" of $\boldsymbol{x}$ which we care most – $p_\theta(\boldsymbol{z}|\boldsymbol{c})$, and leave $p_\theta(\boldsymbol{x}|\boldsymbol{z}, \boldsymbol{c})$ to model other variations including local noise. In this case, CFG will simultaneously sharpen both distributions, considering:

$$p_\theta(\boldsymbol{x}|\boldsymbol{c}) = \sum_{\boldsymbol{z}} \underbrace{p_\theta(\boldsymbol{z}|\boldsymbol{c})}_{\text{mode selection}} \cdot \underbrace{p_\theta(\boldsymbol{x}|\boldsymbol{z}, \boldsymbol{c})}_{\text{image variation}}, \tag{7}$$

where standard diffusion models implicitly learn mode selection step together with generation.

Therefore, a natural solution is to **explicitly** model "mode selection" before applying diffusion steps so that the mode distribution will not be distorted by guidance. In this way, the sampling procedure (Eq. (6)) is modified as two steps: $\boldsymbol{z} \sim p_\theta(\boldsymbol{z}|\boldsymbol{c}), \boldsymbol{x} \sim \tilde{p}_\theta(\boldsymbol{x}|\boldsymbol{z}, \boldsymbol{c})$, where CFG can be applied after $\boldsymbol{z}$ is sampled. From the perspective of score function, we rewrite $\tilde{p}_\theta(\boldsymbol{x}|\boldsymbol{c})$ as $\tilde{p}_\theta(\boldsymbol{x}|\boldsymbol{c}, \boldsymbol{z})$ in Eq. (5):

$$\nabla_{\boldsymbol{x}} \log \tilde{p}_\theta(\boldsymbol{x}|\boldsymbol{c}, \boldsymbol{z}) = \gamma \left[\nabla_{\boldsymbol{x}}\left(\log p_\theta(\boldsymbol{x}|\boldsymbol{c}) + \log p_\theta(\boldsymbol{z}|\boldsymbol{x}, \boldsymbol{c}) - \log p_\theta(\boldsymbol{x})\right)\right] + \nabla_{\boldsymbol{x}} \log p_\theta(\boldsymbol{x}). \tag{8}$$

Compared to standard diffusion process, the highlighted term above pushes the updating direction towards the sampled modes at each step. This ensures diverse generation as long as $p_\theta(\boldsymbol{z}|\boldsymbol{c})$ is diverse.

**A Toy Example**  We visualize the effect of explicitly introducing latent priors using a toy dataset with two main classes, each containing two modes. We compare two models: a standard diffusion model conditioned on the major class ID, and a latent-augmented model incorporating subclass ID as priors. Fig. 3 shows that while the standard diffusion model tends to converge to one mode (subclass) with increased guidance, the latent-augmented model captures all modes, showing the benefit of latent priors for improving diversity under high guidance. In practice, given the challenge of identifying all "modes" in real-world data distribution, we next propose to employ an autoregressive model to universally model various latent modes.

### 3.2  Autoregressive Latent Modeling

To capture the complex distribution of real images, it is clearly impossible to assign classes for each mode. However, it is non-trivial to determine (1) the best representations for modes $\boldsymbol{z}$; (2) the suitable generative model that can model $p_\theta(\boldsymbol{z}|\boldsymbol{c})$. Fortunately, the modes that humans can perceive from an image are largely abstract, and such abstract semantics are easily represented in discrete symbols. For example, we can easily describe content differences through natural language, create composite

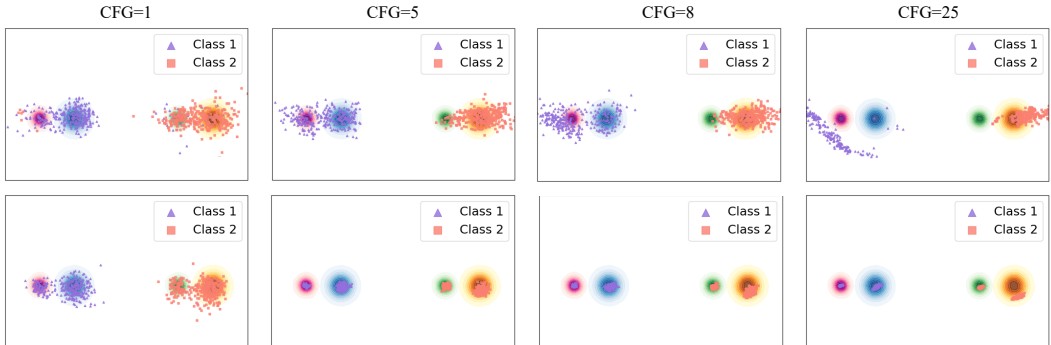

Figure 3: Effect of augmented latents. The first row displays the sampling results from the standard diffusion model, while the second row shows the results from the latent-augmented diffusion models.

images based on spatial locations, and imagine novel visual concepts from experience. Therefore, it is logical to use such **abstract** discrete tokens, i.e., $z = [z_1, \ldots z_N]$. Naturally, the most convenient way to model such distribution is using an autoregressive model $p_\theta(z|c)$. Note that this is distinct from the conventional autoregressive image generation (§ 2), as we only learn such models as "latent modes", where the sampled $z_{1:N}$ are not supposed to reconstruct an image directly. As a result, it eases the modeling difficulty and improves the sampling performance.

In this work, we explore four types of abstract latents: *textual descriptions (text)*, *detection bounding boxes (bbox)*, *object blobs (blob)*, and *visual tokens (voken)*, all of which can be predicted from multi-modal large language models (MLLMs) [Bai et al., 2023, Liu et al., 2024, Ge et al., 2023a] given the condition-image pair $(c, x)$. Each type aims to enrich the mode-to-image correspondence, covering different aspects of image formation. These extracted tokens can either be predicted separately or modeled together with a single autoregressive model. Fig. 4 shows the examples of these four generated latent variables. The methodology for constructing the training dataset for these variables is detailed in Appendix A.

### 3.3 Joint Learning of Autoregressive and Diffusion Models

Similar to other latent variable models like VAEs [Kingma and Welling, 2013], Kaleido can be trained to maximize the evidence lower bound (ELBO) as follows:

$$\max_\theta \log p_\theta(x|c) \geq \mathbb{E}_{z \sim q(z|x,c)} [\underbrace{\log p_\theta(z|c)}_{\mathcal{L}^{AR} \text{ Eq. (1)}} + \underbrace{\log p_\theta(x|z,c)}_{\mathcal{L}^{DM} \text{ Eq. (3)}}] + \mathcal{H}[q(z|x,c)], \qquad (9)$$

where $q$ is the inference model, and $\mathcal{H}(q)$ is the entropy. In this paper, we always assume a fixed inference process (as explained § 3.2). Therefore, the entropy term can be omitted, and we can efficiently sample and store $z$ for the entire dataset before training starts. We illustrate the training pipeline in Fig. 2. Compared to standard diffusion models which typically involves a context encoder and a denoising network, Kaleido integrates the additional autoregressive decoder for modeling the discrete latents. Such decoder uses cross-attention to gather the encoder states at every step, and the final decoder layer states are concatenated with the encoder as the inputs for diffusion. Following common practices, we freeze the context encoder during training, and jointly optimize the autoregressive decoder together with the denoising model. The training objectives (Eq. (9)) is equivalent to the combination of both models, denoted as $\mathcal{L} = \mathcal{L}^{DM} + \eta \cdot \mathcal{L}^{AR}$, with $\eta$ as a hyperparameter for balancing the contributions of the autoregressive and diffusion models in practice.

### 3.4 Interpretable and Controllable Generation

During the inference stage, given the provided textual description, the autoregressive model will first predict the discrete latents before image generation. These latents, being predominantly human-readable, add a layer of interpretability to the image-generation process, allowing humans to observe its internal "thought" process. This transparency also provides users with the flexibility to modify the latents as desired. To incorporate user modification, the altered latents are re-input into the autoregressive decoder to obtain the modified final hidden states. The latent-augmented diffusion model then synthesizes the final image conditioned on the updated representation.

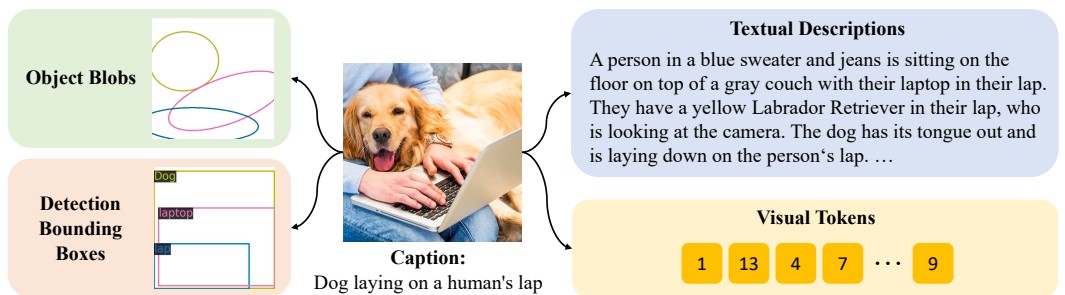

Figure 4: **A Variety of Discrete Tokens.** Original caption: "Dog laying on a human's lap"

# 4 Experiments

## 4.1 Experimental Setups

**Dataset** We validate our approach on both class- and text-conditioned image generation benchmarks. For the former, we use ImageNet [Deng et al., 2009], and we learn the text-to-image models on CC12M [Chang-pinyo et al., 2021], a large image-text pair dataset where each image is accompanied by a descriptive alt-text. All models are trained to synthesize at

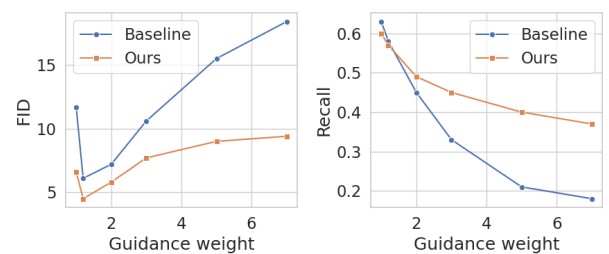

Figure 5: **Comparison with guidance weights.**

$256 \times 256$. We generate all four types of latents as discussed in Appendix A for both datasets.

**Evaluation Metrics** To assess the performance of our models, we employ Fréchet Inception Distance (FID) [Heusel et al., 2017] to capture the overall performance (considering both quality and diversity) of the generated images, and use Recall [Kynkäänniemi et al., 2019] to specifically measure the diversity of the generated images. Furthermore, we employ two additional quantitative assessments of diversity: Mean Similarity Score (MSS) and Vendi scores [Friedman and Dieng]. We use SSCD [Pizzi et al., 2022] as the pretrained feature extractor for calculating both MSS (SSCD) and Vendi (SSCD). Additionally, we utilize DiNOv2 [Oquab et al.] as the feature extractor for Vendi (DiNOv2), based on evidence from Stein et al. [2024] that suggests DiNOv2 provides a richer evaluation of generative models.

**Implementation Details and Baseline** We implement Kaleido with Matryoshka Diffusion Models (MDM) [Gu et al., 2023], a recently proposed approach that generates images directly in the raw pixel space with efficient training. The default MDM consists of a frozen T5-XL [Radford et al., 2021] context encoder and a nested UNet-based denoiser. We initialize the additional autoregressive decoder with the decoder of T5-XL, and make the parameters trainable. The vocabulary is resized to adapt special visual tokens. For fair comparison, we use MDM with the same hyper-parameters as our baseline model, and train both types in almost identical settings on 64 A100 GPUs. Additionally, we compare Kaleido with the Condition Annealed Diffusion Sampler (CADS) [Sadat et al.], a general sampling strategy that enhances the diversity of diffusion models by annealing the conditioning signal during inference. Given that CADS is applicable to different model architectures, we also evaluate CADS integrated with both baseline model MDM (MDM + CADS) and our model (ours + CADS).

## 4.2 Quantitative Results

Fig. 5 quantitatively compares Kaleido with the baseline diffusion models (MDM) with various guidance scales on ImageNet. Both metrics are evaluated with 50K samples against the full training set, where both our models and the baseline use DDPM sampling with 250 steps. Our findings reveal that Kaleido consistently enhances the diversity of samples without compromising their quality across different CFG, evidenced by the general improvement in both FID and Recall. Moreover, while the baseline's FID increases and Recall decreases significantly with higher CFG, Kaleido demonstrates a steadier performance profile.

| Model | FID-50K ↓ | Precision ↑ | Recall ↑ | MSS (SSCD) ↓ | Vendi (SSCD) ↑ | Vendi (DiNOv2) ↑ |
|---|---|---|---|---|---|---|
| MDM | 15.5 | **0.93** | 0.22 | 0.21 | 8.42 | 3.04 |
| MDM + CADS | 10.6 | 0.60 | **0.62** | **0.12** | **9.28** | 4.72 |
| Ours | 9.0 | 0.85 | 0.42 | 0.16 | 8.82 | 3.79 |
| Ours + CADS | **5.9** | 0.76 | 0.52 | **0.12** | 9.21 | **4.83** |

Table 1: **Comparison of quality and diversity on ImageNet.** FID-50K, Precision, and Recall are evaluated on 50K samples, while MSS and Vendi scores assess diversity on $1K \times 10$ samples.

To further investigate, we examine image quality and diversity between Kaleido and baseline models. As shown in Table 1, Kaleido outperforms the MDM + CADS combination in terms of FID-50K and precision, demonstrating that our method more effectively maintains high image quality while generating diverse samples. Furthermore, integrating CADS with our model yields the best FID-50K results. Note that precision cannot accurately evaluate models with diverse outputs since a model producing high-quality but non-diverse samples could artificially achieve high precision [Sadat et al.].

Moreover, we assess the diversity of the generated images using 10K samples. Following CADS, we select $1,000$ random classes from ImageNet and generate 10 samples per class. Table 1 shows that both Kaleido and CADS significantly enhance sample diversity. While CADS achieves better performance in diversity, our model maintains superior image quality. Additionally, the methodologies used in CADS are complementary to ours, suggesting potential benefits from integrating CADS with our Kaleido. In fact, incorporating CADS into our model not only further improves image quality but also improves diversity, achieving the best scores in FID-50K, MSS (SSCD), and Vendi (DiNOv2).

Lastly, we provide visual comparisons for class- and text-conditioned image generation in Fig. 11. Notably, we observe that MDM + CADS fails to generate cats of diverse breeds from the prompt "a cat sleeping on the bed." In contrast, Kaleido can produce images of cats from various breeds with more diverse surrounding environments, showcasing its superior diversity capabilities. This observation contrasts with the trend of diversity scores in Table 1, suggesting that these diversity metrics may not fully capture certain aspects of diversity.

### 4.3 Qualitative Results

**Diversity of Generated Images** We present a comparative analysis of the images generated by Kaleido against baseline models (MDM). Fig. 7 demonstrates the comparison between baseline models and Kaleido on two conditional generation tasks: the class-conditioned image generation and the text-to-image generation. In both tasks, Kaleido consistently produces more diverse images from identical condition (class or textual description) across varying CFG scales. For instance, in the task of class-to-image generation, the baseline diffusion models generate predominantly frontal views of a "husky" at high CFG, while Kaleido produces diverse images depicting huskies in various poses and numbers. A similar improvement in diversity is observed in the text-to-image generation as well, highlighting the robustness of Kaleido in generating diverse images under identical conditions.

**Control from Latent Tokens** We show the efficacy of latent variables in guiding the image generation process in Fig. 6. Fig. 6 demonstrates images generated with different types of latent variables: (a) textual descriptions, (b) object blobs, (c) detection bounding boxes, (d) visual tokens, and (e) combined latents, which integrate textual descriptions, detection bounding boxes and visual tokens. We visualize the generated latents tokens alongside the resulting images, showing how closely the images generated by Kaleido align with the latent tokens. Such alignment is evident in fine-grained visual information – such as object appearance, background, and atmosphere –, spatial location and orientation of different objects, and the stylistic elements of generated images. This alignment confirms that Kaleido can effectively interpret and utilize generated latent variables to guide and refine the image generation process.

**Latent Editing** Fig. 8 showcases the impact of latent editing in image generation. The first row displays images generated using autoregressively produced latent tokens. In the second row, we demonstrate the effect of manual modifications to the textual descriptions: changing "log" to "cobblestones" and "a body of water" to "forest". These changes result in a modified image where a frog is now positioned on cobblestones with a forest background. Additionally, by further augmenting the bounding box of a cup to a different position, we observe that the cup's position in the image changes accordingly, while most other visual elements remain unchanged. The precise control of image characteristics via latent editing underscores Kaleido's flexibility and controllability, offering

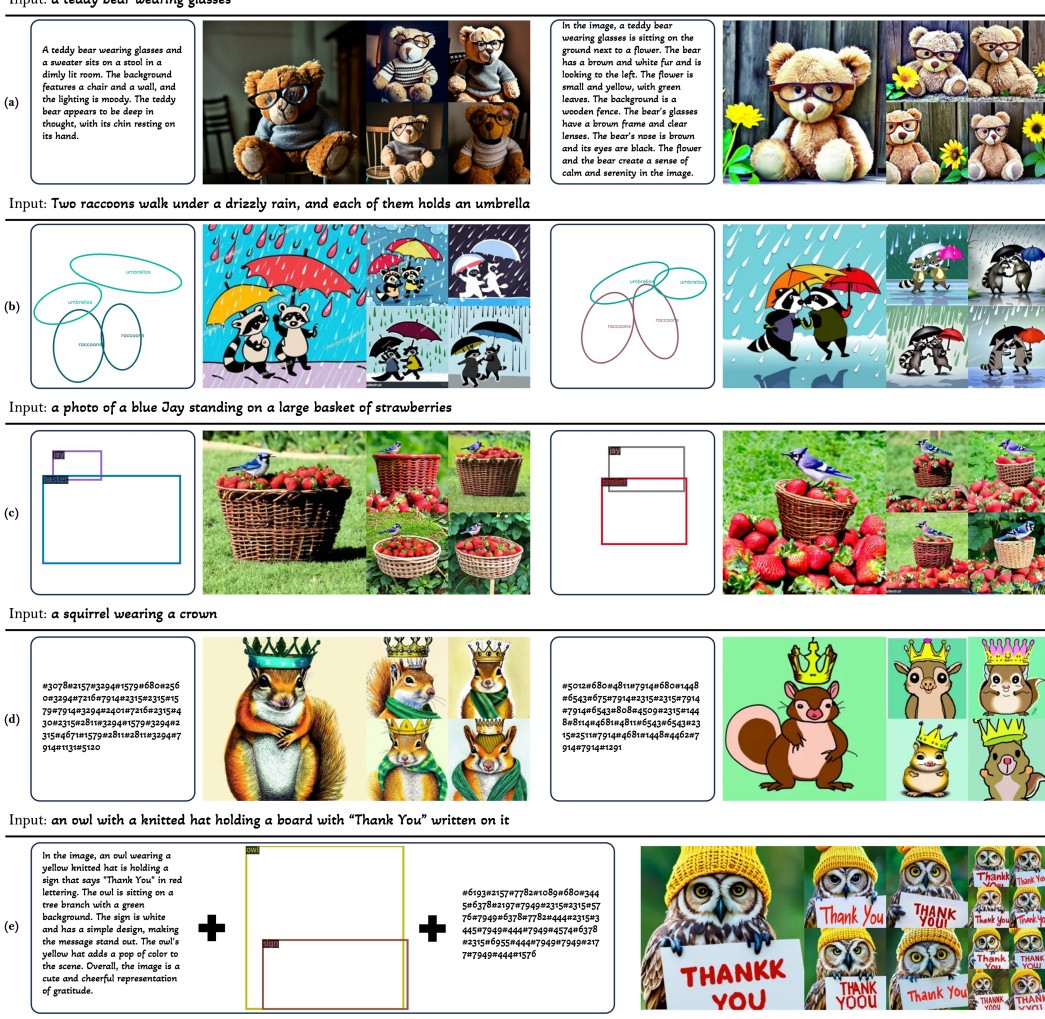

Figure 6: **Example of generation with various latents.**. This figure showcases images generated with different types of latents: (a) textual descriptions, (b) object blobs, (c) detection bounding boxes, (d) visual tokens, and (e) combined latents (textual descriptions + detection bounding boxes + visual tokens). Each row shows two sets of generated images sampled with one type of latents. Each set displays a visualization of the generated latents tokens (left) and a collage of images (right) sampled using the same latent tokens but different noises. The image tokens capture visual details difficult to convey through text, such as artistic style.

a powerful interactive interface for users to customize the generated images. Furthermore, the high fidelity of the re-generated images to their original versions indicates Kaleido's potential for applications requiring personalization or customizations.

## 5 Related Work

**Augmenting Diffusion Models**   Various enhancements have been proposed to improve the versatility and controllability of diffusion models with augmented latents. Innovations such as Diffusion AE [Preechakul et al., 2022] integrates diffusion models with a learnable encoder that extracts high-level semantics and enables the diffusion model to add details directly in image space. Further efforts have focused on incorporating specific control signals, such as bounding boxes, layout, and segmentation masks to guide and control the image generation process. [Balaji et al., 2022, Li et al., 2023, Zheng et al., 2023a, Hu et al., 2023]. Recently, BlobGen [Nie et al., 2024] proposes to ground existing text-to-image diffusion models on object blobs – tilted ellipses that capture spatial details of the objects – for compositional generation. While these approaches improve the models' capacity to

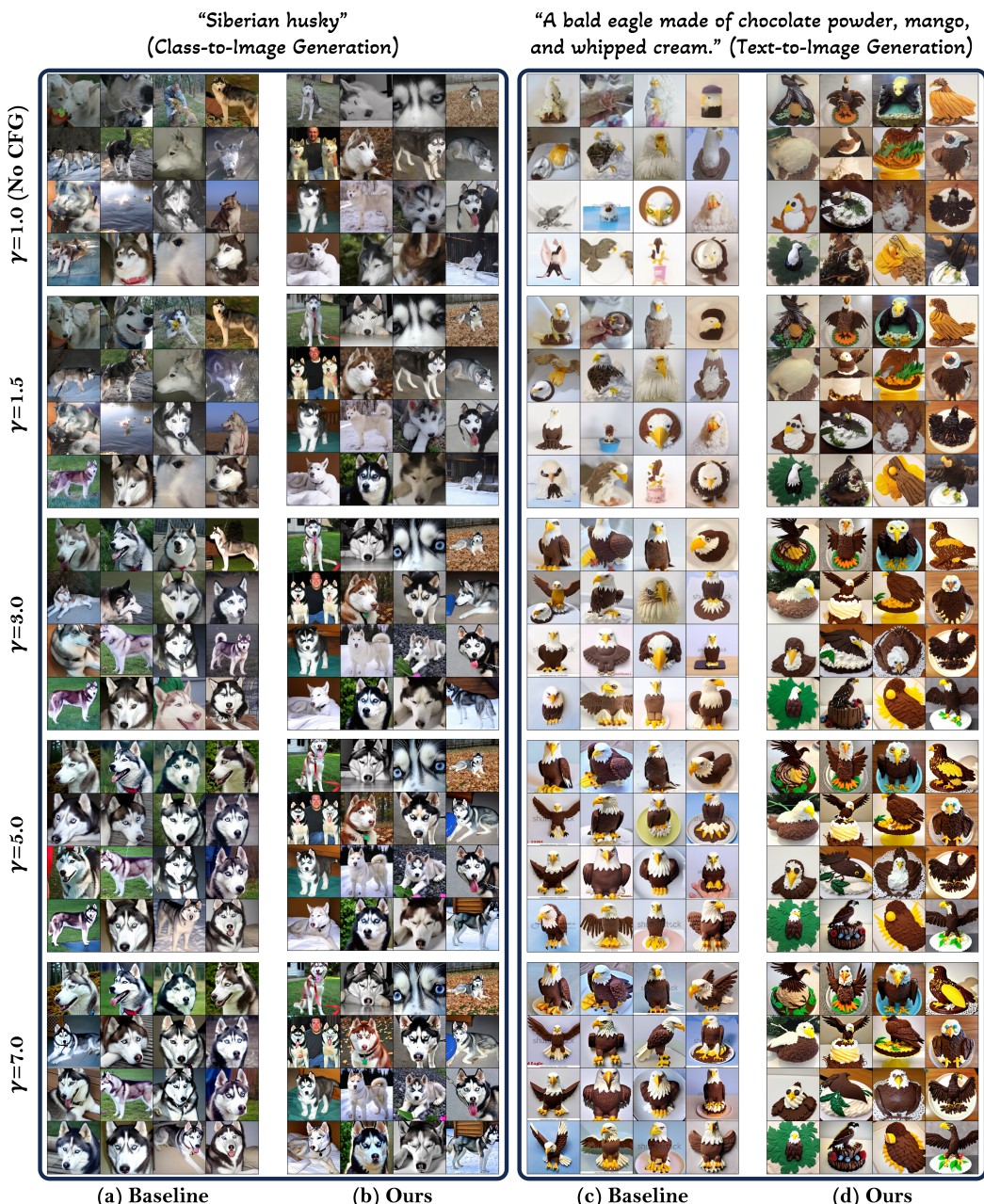

Figure 7: **Diversity comparison to standard diffusion model.** Images sampled under varying CFG scales ($\gamma$). Panels (a) and (c) display images from the baseline models, while panels (b) and (d) show images from Kaleido. From top to bottom, as the CFG increases, the standard diffusion models exhibit reduced diversity, while Kaleido consistently maintains diversity across guidance scales.

adhere to specified spatial layouts, they often necessitate modifications to the attention mechanism, potentially limiting their generality. In contrast, our method enhances the generative capabilities of diffusion models without altering the model architecture.

**Connecting Diffusion Models with LLMs** The remarkable success of Large Language Models (LLMs) and diffusion models has spurred interest in connecting these models, aiming to leverage the capabilities of LLMs in understanding and generating complex data and combine it with the powerful image synthesis capabilities of diffusion models [Ge et al., 2023b, Zheng et al., 2023b, Sun et al., 2023]. Ge et al. [2023b], Zheng et al. [2023b] propose image tokenizers that encodes images into

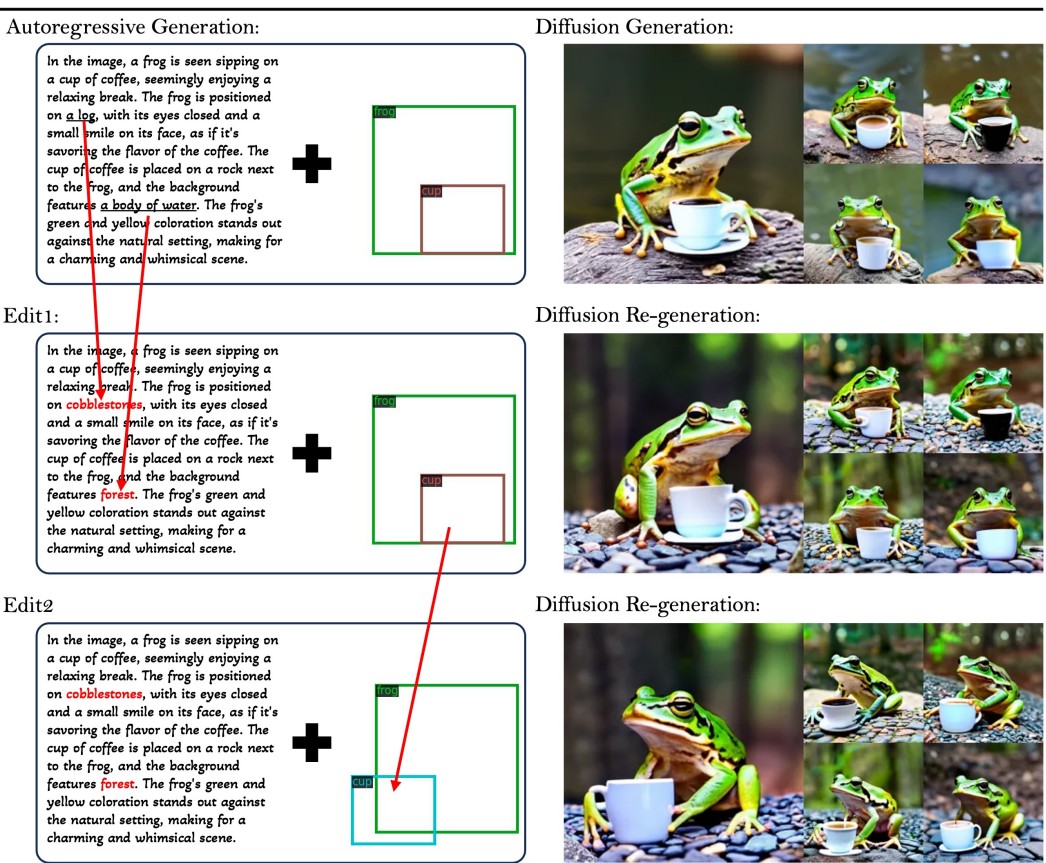

Figure 8: **Effect of sequential latent editing.** The top row displays images generated with autoregressively produced latent tokens. The middle row shows the re-generated images after applying latent editing to the textural description, and the bottom row presents re-generated images after further edits to the bounding box, showing the impact of step-by-step latent editing.

visual tokens, enabling multimodal language modeling. This line of work focuses on empowering LLM with image generation ability by aligning its output embedding space with the pre-trained diffusion models. Our work leverages the LLMs' robust capabilities in textural understanding and generation to model the generation of abstract latents from the original text. These latents are then integrated with latent-augmented diffusion model, enabling a more interpretable and diverse image generation process.

Our approach also distinguishes itself from the re-captioning method introduced in DALL-E 3 [Betker et al., 2023]. Unlike re-captioning, which typically replaces the original captions with more descriptive captions, our method retains the original condition and supplements it with latent variables of various forms (beyond textual captions like bbox, blob and "vokens"). The sampled latents serves as a unifying interface for various types of inputs, and introduce diversity compared to recaptioning where no sampling is involved at inference time.

# 6   Conclusion

In this work, we address the challenge of improving sample diversity under high CFG in diffusion models. We introduce Kaleido Diffusion, which combines an autoregressive prior with a latent-augmented diffusion model. Results show Kaleido increases diversity without compromising quality, even at high CFG. With human interpretable latent tokens, Kaleido offers an explainable mechanism behind the image generation process and provides a fine-grained editable interface, enabling precise user control over the generated images.

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

# Appendix

## A   Auto-regressive Latent Modeling

In this work, we explore four types of abstract latents, including textual descriptions (text), detection bounding boxes (bbox), object blobs (blob), and visual tokens (voken). Each type is designed to enrich the mode-to-image correspondence, covering different aspects of image formation. Examples of these abstract latents are illustrated in Fig. 4. In the following paragraphs, we detail the methodology employed in constructing the training dataset for these abstract latents. Additionally, Fig. 9 outlines the pipeline for the step-by-step generation of these abstract latents.

**Textual descriptions**   Typical text-image datasets often provide captions that fail to fully capture the details of the image. For instance, as shown in Fig. 4, the original caption "Dog laying on a human's lap" omits crucial details such as the presence of "laptop", which is essential for accurate image

---

**Textual descriptions (caption)**

---

Original caption: {}
Using the information provided in the caption above, Please provide a detailed description of the image in 50-80 words, incorporating relevant information from the caption and expanding on the visual elements:

- Include names, objects, events, and locations mentioned in the caption
- Do not include placeholders like <PERSON> in the caption
- Describe people, characters, animals, and notable entities
- Mention the setting, background, and overall environment
- Note colors, lighting, composition, and style aspects
- Refer to any text, symbols, or logos in the image

Combine caption details with your observations to create a comprehensive description of the key elements and overall scene, focusing on the most salient aspects of the image.

---

**Textual descriptions (label)**

---

Object labels: {}
Using the provided object labels, generate a detailed description of the image in 50-80 words, incorporating the relevant information about prominent objects identified and expanding on the visual elements:

- Describe each labeled object, including its size, shape, and placement in the scene
- Depict relations between objects, such as proximity or arrangement
- Highlight interactions or functions implied by the objects
- Describe the surrounding environment or context that complements the labeled objects
- Any notable features or characteristics of the objects, like color, texture, or design elements
- Describe people, characters, animals, and notable entities
- Mention the setting, background, and overall environment
- Note colors, lighting, composition, and style aspects
- Refer to any text, symbols, or logos in the image

Craft a comprehensive depiction of the scene based on the identified objects, utilizing both the labels and contextual observations to enrich the description.

---

**Captions with grounding**

---

Generate the caption in English with grounding:

---

Table 2: The instruction for prompting Qwen-VL to generate detailed textual descriptions and captions with grounding.

generation. To address this, we employ detailed textual descriptions as latent variables. These textual descriptions supplement the original captions by providing additional information that might be missing from the original captions. Specifically, we leverage Qwen-VL-Chat [Bai et al., 2023], a large visual language model, designed for effective instruction-following across a variety of multimodal tasks. We instruct Qwen-VL-Chat to produce a detailed textual description given the original caption and corresponding image. The specific instructions used for generating the textual descriptions are detailed in Table 2 under the section *Textual descriptions (caption)*. Fig. 4 shows an example of the generated detailed textual description that provides a more comprehensive depiction of the scenes than the original captions, thus allowing for a richer image generation.

Additionally, for the ImageNet dataset, which consists of label-image pairs for class-to-image generation, we instruct Qwen-VL-Chat to generate detailed descriptions based on the class label and corresponding image. The instruction for this procedure is similarly documented in Table 2 under the section *Textual descriptions (label)*.

**Detection bounding boxes**   The spatial location of objects within an image is also crucial information for accurate representation of the image, yet such information is typically absent in textual descriptions. To incorporate this spatial information into the image generation process, we use detection bounding boxes as one type of abstract latents. Specifically, we use Qwen-VL [Bai et al., 2023] to prompt the model to "Generate the caption in English with grounding:". This approach results in captions where the spatial locations of objects are explicitly annotated within the text. For instance, as shown in Fig. 4, the caption with grounding for this example is: "Dog (1, 33, 995, 995) resting head on owner's lap (1, 630, 785, 998) while they work on a laptop (39, 336, 999, 972)." Each bounding box is described in a string format "$x_1, y_1, x_2, y_2$", where $x_1, y_1$ and $x_2, y_2$ are the coordinates of the top-left and bottom-right corner, respectively. All the coordinates are normalized to a $[0, 1000]$ range. The coordinates string is treated as part of the text, obviating the need for an additional positional vocabulary.

**Object Blobs**   Inspired by Nie et al. [2024], we utilize object blobs as the abstract latents that contain more advanced spatial information. An object blob is defined as a tilted ellipse that specifies the position, size, and orientation of an object within an image. Specifically, a blob is represented as "($x_c$, $y_c$, $r_{major}$, $r_{minor}$, $\theta$)" where $(x_c, y_c)$ denotes the center point of the ellipse, $r_{major}$ and $r_{minor}$ are the radii of its semi-major and semi-minor axes, respectively, and $\theta \in [0, 180)$ denotes the orientation angle of the ellipse. To extract the blobs for meaningful objects, we leverage the results from bounding box detection and employ SAM [Kirillov et al., 2023] to generate the segmentation maps using the bounding boxes as prompts. Subsequently, an ellipse fitting algorithm is applied to these segmentation maps to determine the blob parameters for each identified object. This method allows for a more precise representation of objects' spatial characteristics, thus improving the integration of spatial and structural information within the image generation process.

**Visual Tokens**   Representing images via discrete visual tokens, especially using technologies like Vector Quantized Variational Autoencoder (VQ-VAE) [Van Den Oord et al., 2017], has become a prevalent technique in generative modeling due to its ability to encode high-dimensional image data into a more manageable, discrete space. In this work, we utilize SEED [Ge et al., 2023b], a VQ-based image tokenizer, to encode an image into a sequence of abstract discrete image tokens. These tokens encapsulate high-level semantic information of the visual elements in the image, serving as potent latent variables for guiding the diffusion model. The visual tokens are concatenated with the delimiter "#", forming a sequence of visual tokens represented as "$I_1\#I_2\#...\#I_{32}$", where each "$I_i$" denotes the image token id.

# B   Implementation Details

## B.1   Architecture

In this paper, we use the following NestedUNet architecture proposed in Gu et al. [2023] to implement the denoising model. The total number of parameters is about $500M$. For the autoregressive prior, we employ T5-XL [Raffel et al., 2020] for all experiments regardless of the input latent types. Both the denoiser and T5 decoder receive gradients and are trained end-to-end.

```
config:
    resolutions=[256,128,64]
    resolution_channels=[64,128,256]
    inner_config:
        resolutions=[64,32,16]
        resolution_channels=[256,512,768]
        num_res_blocks=[2,2,2]
        num_attn_layers_per_block=[0,1,5]
        num_heads=8,
        schedule='cosine'
    num_res_blocks=[2,2,1]
    num_attn_layers_per_block=[0,0,0]
    schedule='cosine-shift4'
    emb_channels=1024,
    num_lm_attn_layers=2,
    lm_feature_projected_channels=1024
```

## B.2 Training

For all experiments, we share all the following training parameters for both the baseline model and the proposed Kaleido Diffusion.

```
default training config:
    batch_size=512
    num_updates=400_000
    optimizer='adam'
    adam_beta1=0.9
    adam_beta2=0.99
    adam_eps=1.e-8
    learning_rate=1e-4
    learning_rate_warmup_steps=10_000
    weight_decay=0.0
    gradient_clip_norm=2.0
    ema_decay=0.9999
    mixed_precision_training=bp16
```

All experiments are performed on $64$ A100 GPUs which takes roughly 2 weeks for training 400k steps for both ImageNet and CC12M datasets. For text-to-image models, we perform an additional 400k steps progressive training at $64 \times 64$ resolution, while we train the entire model from scratch directly at $256 \times 256$ for ImageNet. Due to the memory cost of the T5-decoder, we can only fit $4 \sim 8$ images per GPU, causing at least $\times 3$ slower training compared to the original MDM models.

## B.3 Learned Models

To demonstrate the effectiveness of various latents, we train our model with 5 types including *text*, *bbox*, *blob*, *voken*, and *combined* for text-to-image generation. For *combined* setting, we use the autoregressive model to predict

$$combined = text \mid bbox \mid voken$$

in a sequential way such that the latter latents will be controlled by earlier latents. We also trained models on ImageNet using *combined* latents for quantitative comparison.

## C   Limitations

**Training Complexity:**   The enhanced diffusion model may require more complex and extended training processes compared to standard models. This could lead to increased computational costs and longer development times, potentially limiting accessibility for smaller organizations or individual researchers.

**Difficulty in Finding Optimal Latents:** Identifying the most effective latent variables to achieve the desired output diversity can be challenging. This process might involve extensive experimentation and fine-tuning, which can be time-consuming and resource-intensive. Additionally, covering a broader range of modes, such as depth and semantic maps, adds another layer of complexity to the model development, requiring sophisticated techniques to integrate these diverse forms of data effectively.

**Memory Usage:** The improved diffusion model, with its increased output diversity, might demand higher memory usage due to the integration of the heavy language models. However, potential strategies such as partial training or joint training with LLMs could be explored to mitigate this issue. These methods could help distribute the computational load more effectively and reduce the memory footprint during the training process.

## D   Impact Statement

The proposed method to enhance diffusion models and increase output diversity has significant social implications. By advancing the diversity and accuracy of generated outputs, this technology can be leveraged in various fields such as art, media, and content creation, providing more inclusive and representative outputs that reflect a broader spectrum of human experiences and creativity. Moreover, in areas like healthcare and education, diverse and precise models can lead to more personalized and effective solutions, addressing the unique needs of individuals and communities. This innovation also promotes ethical AI practices by reducing biases in model outputs, fostering a more equitable digital landscape. Ultimately, the enhanced diffusion models will contribute to the democratization of AI, making sophisticated tools accessible to a wider range of users and applications, thereby driving societal progress and innovation.

## E   Color Cluster as Additional Latent

We have included an alternative approach that constructs latent tokens without relying on additional knowledge. Specifically, we train a model using color clusters as latent tokens. For each color channel (R, G, and B) within the range of $0 - 255$, we equally segment it into eight clusters, resulting in a total of $8 \times 8 \times 8 = 512$ color clusters. Given an image, we resize it to 4x4 pixels and assign a color cluster ID to each pixel based on its RGB value. The image is then encoded into a sequence of color cluster IDs (e.g., "$C_1\#C_2\#...\#C_{512}$"), with each $C_i$ representing a color cluster ID. This sequence serves as the condition for training our Kaleido diffusion.

In Fig. 10, we showcase images generated using color clusters as latent tokens on ImageNet. Our results demonstrate that, compared to the baseline MDM, our Kaleido diffusion can generate more diverse images with latent tokens derived purely from color clustering. This highlights that Kaleido diffusion's capability to generate diverse images is independent of distilled external knowledge, confirming that our approach can produce varied images without the aid of any other pre-trained models.

## F   Qualitative Comparison with CADS

Fig. 11 shows the visual comparisons for class- and text-conditioned image generation with CADS.

## G   Additional Results

We show additional results randomly sampled from our models. For all results including the baseline model, we use DDPM sampling with 250 steps.

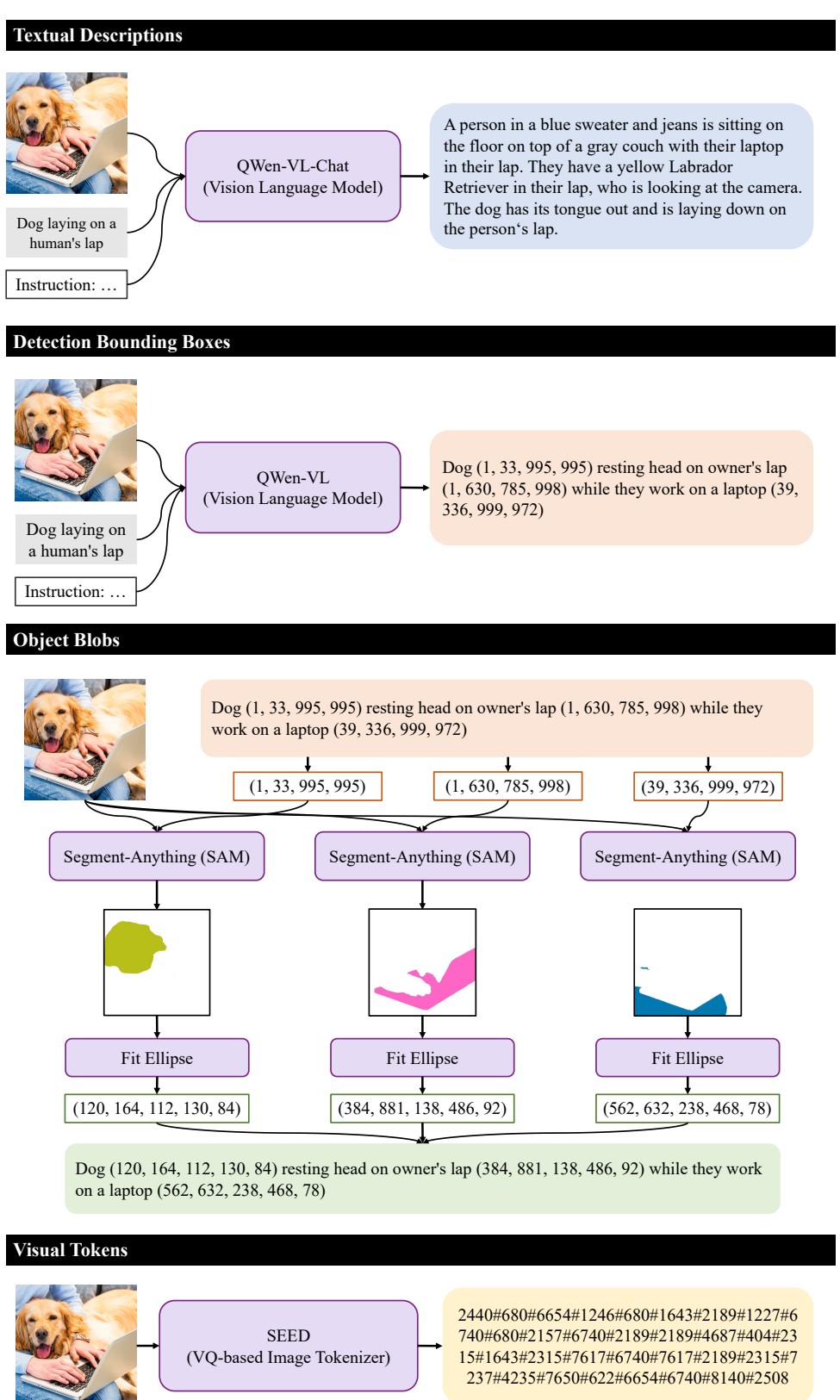

Figure 9: Pipeline for generating various discrete latents.

Baseline MDM on ImageNet Kaleido-MDM with Color Clusters as **Latents** (Ours)

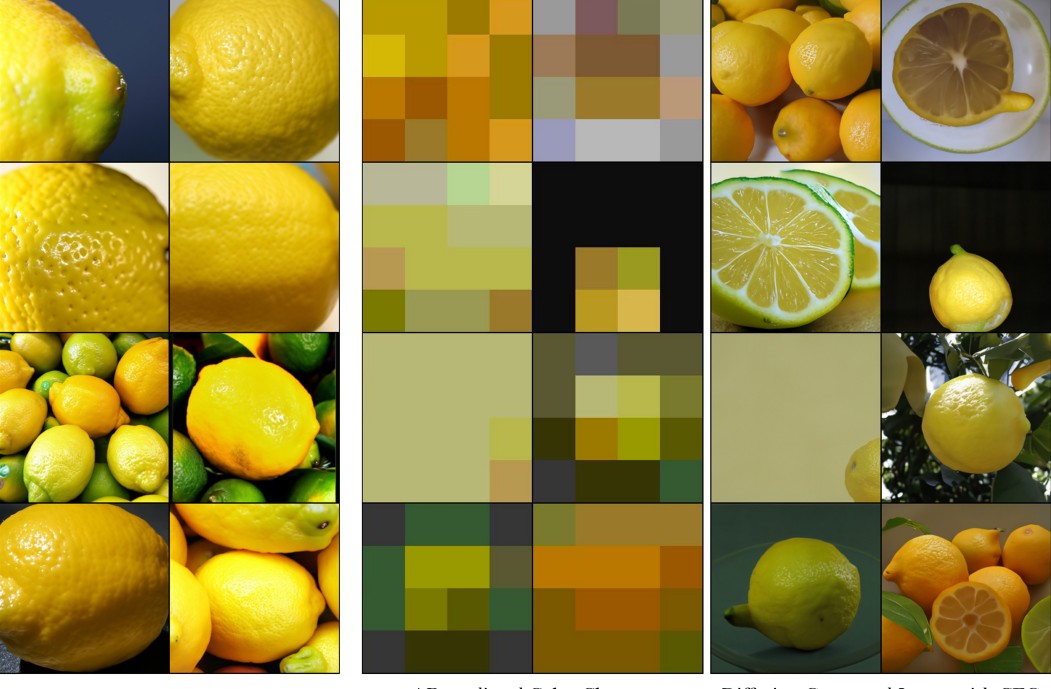

AR predicted Color Clusters Diffusion Generated Image with CFG

Figure 10: Kaleido-MDM with color clusters as latents on ImageNet (class: *lemon*)

MDM-DDPM  MDM-CADS  Kaleido-DDPM  Kaleido-CADS  MDM-DDPM  MDM-CADS  Kaleido-DDPM  Kaleido-CADS

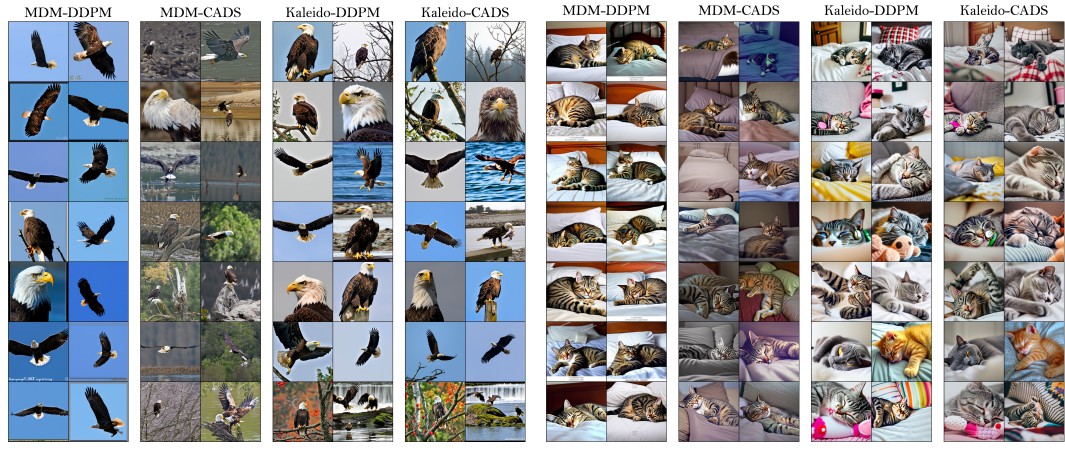

(a) Class-conditioned Image Generation (class: *Bald Eagle*).

(b) Text-conditioned Image Generation (prompt: *a cat sleeping on the bed*).

Figure 11: Qualitative Comparison with CADS

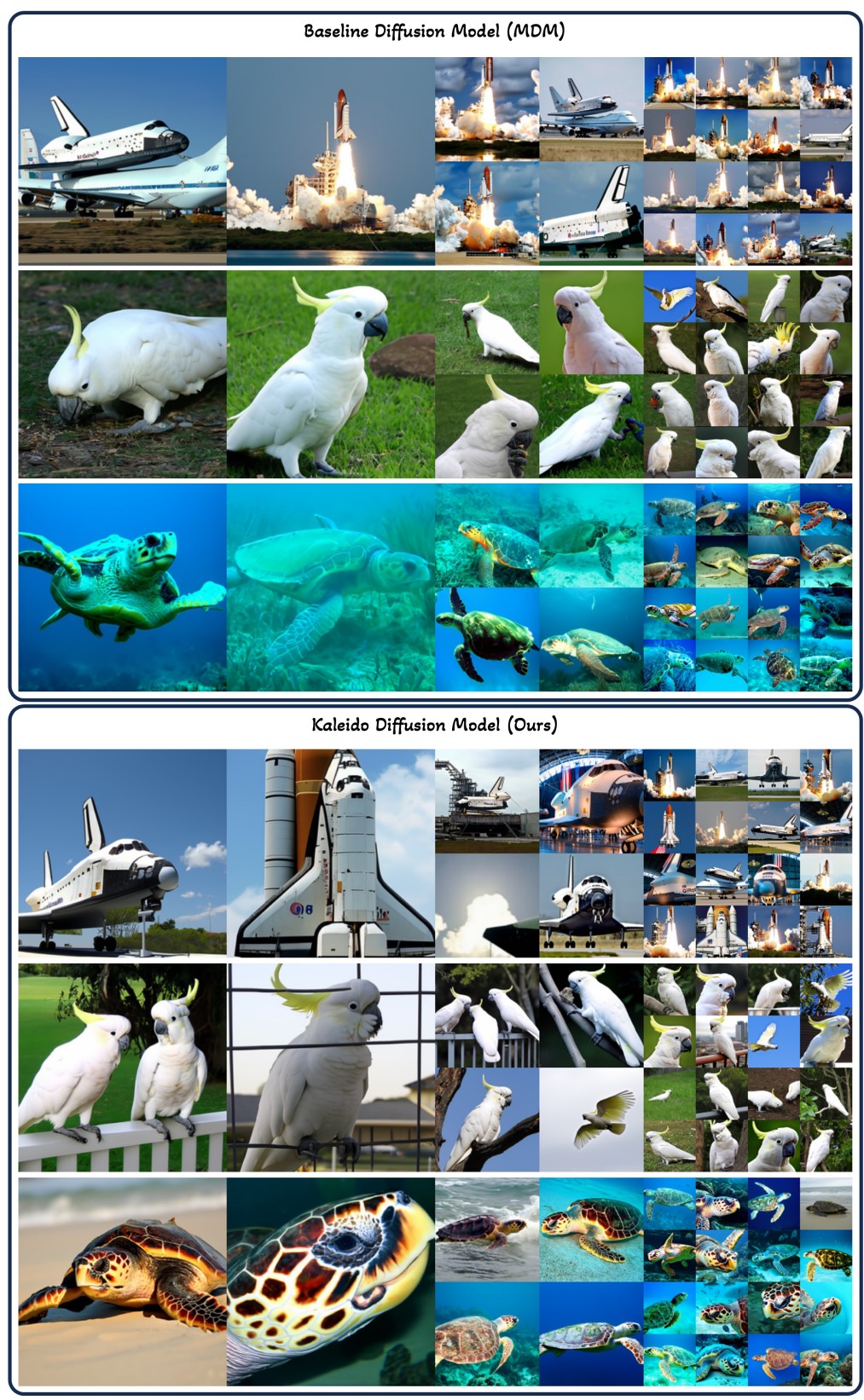

Figure 12: Uncurated samples for both the baseline (MDM) and the proposed Kaleido Diffusion on ImageNet $256 \times 256$. The guidance scale is set $4.0$.

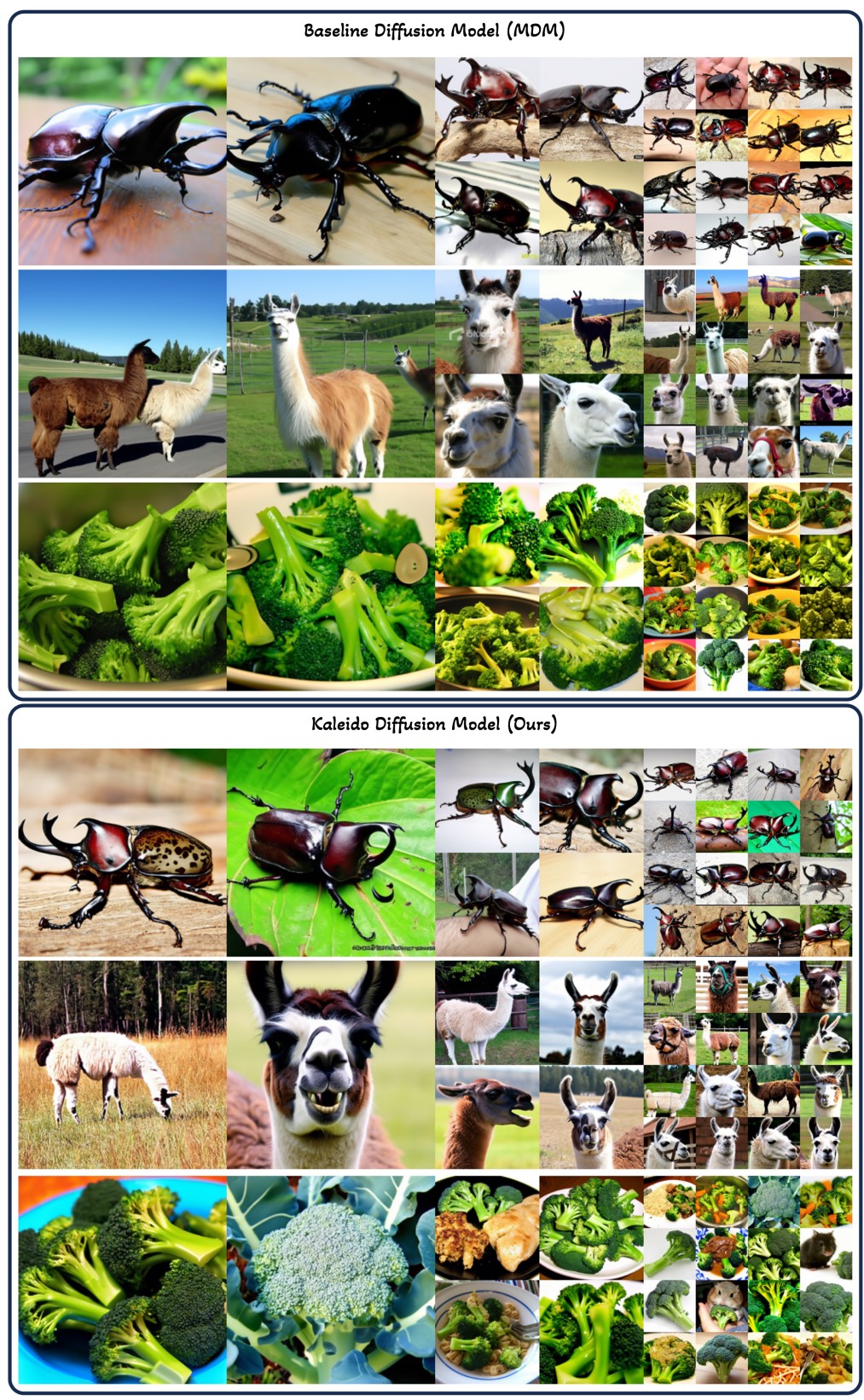

Figure 13: Uncurated samples for both the baseline (MDM) and the proposed Kaleido Diffusion on ImageNet $256 \times 256$. The guidance scale is set $4.0$.

a teddy bear wearing glasses

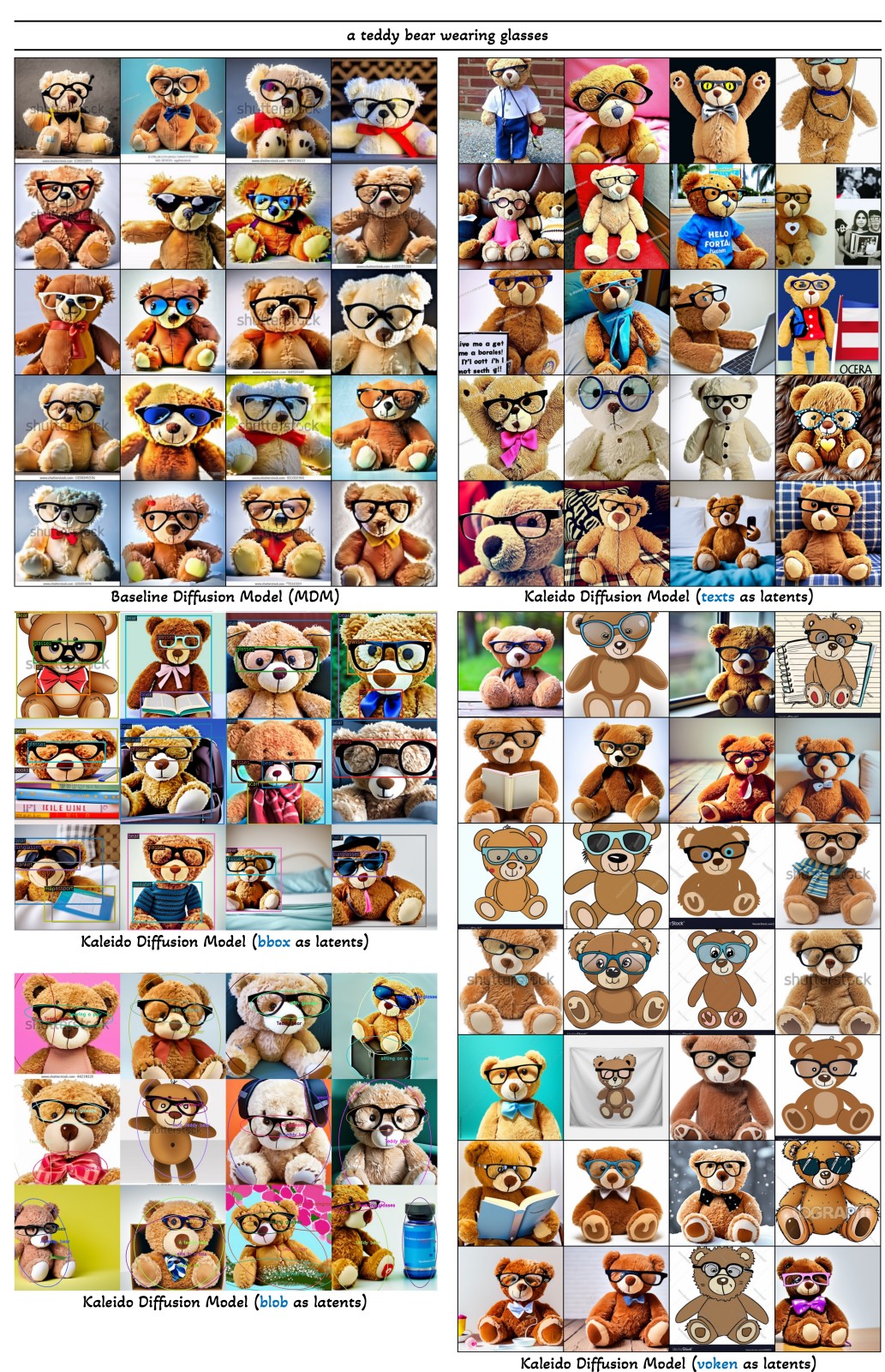

Figure 14: Uncurated samples for both the baseline (MDM) and the proposed Kaleido Diffusion (using *text,bbox,blob,voken* latents) on CC12M 256 × 256 given the same condition. We visualize the generated bounding-boxes and blobs for the ease of visualization. The guidance scale is set 7.0.

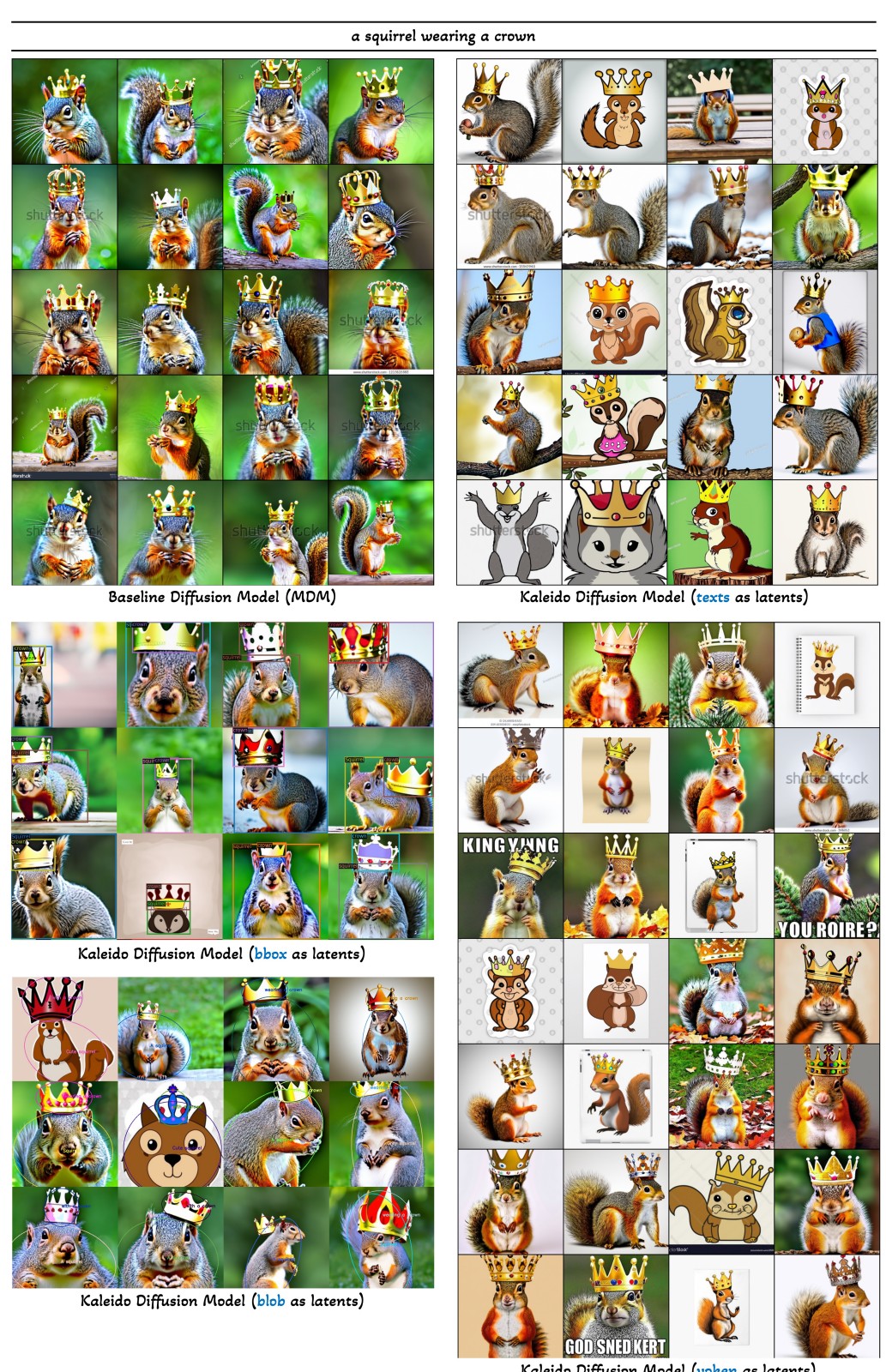

Figure 15: Uncurated samples for both the baseline (MDM) and the proposed Kaleido Diffusion (using *text,bbox,blob,voken* latents) on CC12M $256 \times 256$ given the same condition. We visualize the generated bounding-boxes and blobs for the ease of visualization. The guidance scale is set 7.0.

Input: **A dog is reading a thick book**

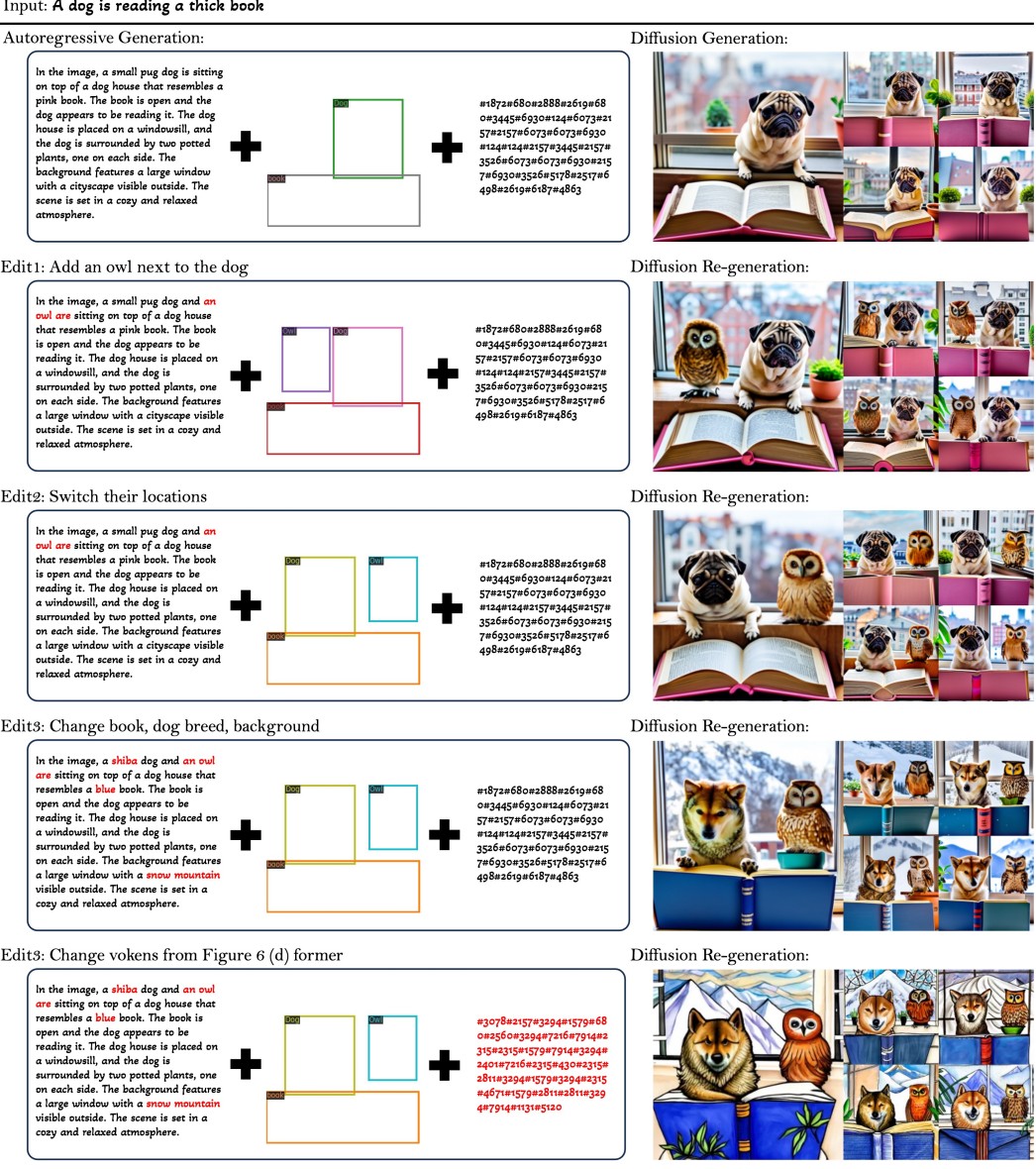

Figure 16: Interactive example of editing the generation process by manipulating the autoregressive predicted latents. The top row displays images generated using autoregressively produced latent tokens, and the subsequent rows show the images re-generated after applying editing on the latents. The guidance scale is set 7.0.

