# OpenReview forum: "Kaleido Diffusion: Improving Conditional Diffusion Models with Autoregressive Latent Modeling"
_NeurIPS.cc/2024/Conference — NeurIPS 2024 poster_

### Official Review · Reviewer_PSab · 2024-07-10

**Soundness:** 3
**Presentation:** 3
**Contribution:** 3
**Rating:** 6
**Confidence:** 4

**Summary:**

Diffusion models generate high-quality images from text but often lack diversity, especially with high classifier-free guidance. Kaleido addresses this by using autoregressive latent priors, which generate diverse latent variables from captions. It enriches input conditions, resulting in more diverse outputs while maintaining quality. Experiments confirm Kaleido's effectiveness in increasing image diversity and adherence to guidance.

**Strengths:**

1. Due to the tendency of samples to converge towards the direction indicated by the condition under high CFG settings, the approach of generating abstractions first to expand the diversity of conditions and then proceeding with generation seems a very reasonable approach. Moreover, these abstractions are controllable, enhancing interpretability and customization possibilities, which is also favorable.
2. The qualitative results also appear promising.
3. The approach of constructing fine-grained prior via autoregression is novel to my knowledge

**Weaknesses:**

1. The formulation in Section 3.1 seems somewhat unintuitive. From my perspective, both text descriptions and the additional autoregressive priors you construct are forms of condition signals for diffusion models. Therefore, I do not see a compelling reason to modify or complicate the original classifier-free guidance formulation. Why not simply regard your method as an extension of the condition signal for classifier-free guidance, while still adhering to the existing CFG formulation?
2. The quantitative results are somewhat limited, with MDM being the only baseline. More comparisons with state-of-the-art diffusion models are encouraged.

**Questions:**

1. What is the motivation behind introducing visual tokens in the autoregressive prior? From the overall narrative of your paper, it seems that the autoregressive prior is intended to provide more customized control for users. However, visual tokens are essentially uninterpretable, making them unusable for user control.
2. The generation process involves an autoregressive process at the beginning. How does this process affect the efficiency of your method?

**Limitations:**

More results on the efficiency and quantitative performance of your method are encouraged.

---

> ### Author Rebuttal · Authors · 2024-08-07
>
> ## **W1: Explanation of the formula in Sec. 3.1**
>
> Thank you for your valuable feedback.
> - In our practical implementation, we indeed adhere to the existing classifier-free guidance (CFG) formulation. Both text descriptions and our introduced autoregressive priors indeed serve as conditioning signals for the diffusion models. Our sampling method can indeed be seen as an extension of the conditioning signal in CFG, specifically through the incorporation of autoregressive priors.
> - The formulation presented in Section 3.1 is designed to elucidate how these autoregressive priors effectively address diversity issues when operating under high CFG settings. Therefore, this is indeed a mathematical motivation rather than an actual implementation.
>
> We will ensure these points are more clearly articulated in the revised version.
>
>
> ## **W2: More Quantitative Results and comparison with sota diffusion models / L1: More quantitative Results:**
>
> Thank you for your feedback!
> - Our proposed framework integrates an autoregressive prior with a diffusion model to enhance image generation. Theoretically, this approach is compatible with a variety of backbone diffusion models. In this paper, we focused on comparing our model with the MDM baseline in pixel space to achieve a fair comparison and clearly demonstrate the specific improvements our framework offers. As part of future experiments, we will also apply our approach to other model types, such as latent diffusion and flow matching.
> - **Please refer to the general response to all reviewers for detailed results and a discussion** of the comparison between our method and CADS—a state-of-the-art, training-free approach that enables diverse generation from diffusion models.
>
>
> ## **Q1: Motivation Behind Introducing Visual Tokens**
>
> - The primary motivation for incorporating the autoregressive prior (i.e., latent tokens) is to explicitly model the mode selection distribution $p(z \mid c)$, enabling the generation of diverse image samples from the same condition, even under high CFG settings. This approach also introduces an explainable and editable mechanism into the image generation process. Our experiments demonstrate that visual tokens (vokens) complement text tokens and are particularly effective at capturing visual details difficult to convey through text, such as artistic style, as shown in Fig. 6 of the main paper.
> - We would also like to clarify that vokens are **interpretable** and **can be manipulated by users for image editing**. The training dataset of visual tokens is constructed by encoding the images into discrete image tokens using SEED [1], a VQ-VAE-based image tokenizer [2] (L748-756). Therefore, we can interpret the generated visual tokens by decoding them back into images using the corresponding image de-tokenizer (i.e., decoder). User control over the image editing process can be achieved by replacing the vokens with different visual tokens. In Fig. R.6 of our rebuttal PDF, we show that by replacing the visual tokens, users can alter the style and characteristics of the re-generated image.
>
> [1] Planting a Seed of Vision in Large Language Models
> [2] Neural Discrete Representation Learning
>
> ## **Q2: Model Efficiency / L1: Results on Efficiency**
>
> AR sampling is performed only once before the diffusion steps, making its running cost negligible.
> - Specifically, in text-to-image settings for generating 256x256 images with classifier-free guidance (batch-size=32), the AR part takes **6 seconds** while the MDM part takes **110 seconds** on a single H100 GPU when using DDPM with 250 steps, which dominates the majority of the time.

---

> ### Author Response · Authors · 2024-08-12
> **Discussion Period**
>
> Dear Reviewer,
>
> As the discussion period deadline nears, we would greatly appreciate it if you could review our rebuttal and share any further feedback. If there are still concerns, we would greatly appreciate a list of specific changes you would need to reconsider your rating.
>
> Thank you for your time and consideration.
>
> Best regards,

---

> > ### Comment · Reviewer_PSab · 2024-08-13
> >
> > Thanks for your clarification. I would like to keep my positive rating.

---

> > > ### Author Response · Authors · 2024-08-13
> > >
> > > Thank you for your valuable feedback! We hope our responses have clarified your concerns!

---

### Official Review · Reviewer_6rHP · 2024-07-11

**Soundness:** 3
**Presentation:** 3
**Contribution:** 2
**Rating:** 6
**Confidence:** 4

**Summary:**

This paper introduced Kaleido Diffusion which leverages an autoregressive model to first model the latent mode and then generate latents based on the sampled mode. The proposed method is reasonable. The authors explain the insight from a classifier-free guidance perspective. Several experimental results can support the authors' claim.

In addition, the paper with the same contents is published as a workshop paper in ICML24.

**Strengths:**

- This paper investigates how to improve the diversity of generated images with additional mode controls, which is interesting.
- The writing is clear and the mathematical explanation seems reasonable.

**Weaknesses:**

- The quantitative results are very limited (only Figure 5).
- What is the context extractor (MLLM) used in the experiments? Although the authors claimed the mode selection is the major contribution of this work, the additionally enrolled pseudo labels perhaps also contribute to the performance improvements.
- I am not convinced that the proposed approach is closely connected to the CFG explanation. Several previous studies, eg, ControlNet (ICCV23) and MaskComp (ICML24) have proven that dense controls will improve image quality. The proposed method is more like distilling knowledge from the pre-trained MLLM to obtain more detailed semantic information about the original dataset.
- It would be better to discuss more implementation details to enhance reproducibility.

**Questions:**

- The dataset section claimed the usage of ImageNet and CC12M. However, the results of ImageNet are missing in the entire paper.
- Which dataset was used for the results reported in Fig 5?
- Can you quantitatively measure the diversity?

**Limitations:**

Yes

---

> ### Author Rebuttal · Authors · 2024-08-07
>
> ## **W2: Use of MLLM/context extractor; Introduction of pseudo labels**
>
> The context (latent) extractor is employed to extract different types of abstract latents given the condition-image pair (Sec. 3.2). In practice, for different types of latents, we utilize different methods as the context (latent) extractors. The construction of the training dataset for these latent tokens is explained in Appendix A.
>
> We appreciate the reviewer's concern regarding the potential impact of pseudo labels on our results. However, it is crucial to clarify that the improvements in our work are not solely due to the introduction of pseudo labels. Simply introducing pseudo labels, without modeling them as latents, would not alone lead to the observed improvements in image diversity.
>
> - To illustrate this point, we have conducted additional training with the MDM (baseline) model, using the generated synthetic textual description (i.e., pseudo tokens) as the condition for training. In Fig.R.3 of the rebuttal PDF, we demonstrate that merely using pseudo labels in this manner does not enhance the diversity, underscoring that the effectiveness of pseudo labels depends significantly on the thoughtful integration into the modeling process. For both models, all images are generated using DDPM with 250 steps and a CFG of 5.
>
> ----
>
> ## **W3: The CFG explanation & knowledge distillation**
>
> We appreciate the reviewer’s reference to prior studies, such as ControlNet [1] and MaskComp [2]. While these contributions are significant, they are not designed to improve image diversity given the same input condition $c$.
> - In ControlNet, the text condition and dense control (e.g., canny edge) together form the user input $c$. Although using different additional controls (considered as different modes, $z′$ can produce diverse outputs, this approach differs from our focus. Our objective is to enhance diversity without requiring additional user-provided information.
> - Similarly, MaskComp [2] addresses the object completion task through iterative generation and mask segmentation but does not prioritize diverse generation of partial masks.
>
> Our design is closely connected to the CFG explanation:
> - In a standard diffusion model, increasing CFG sharpens the conditional distribution $p_\theta(c \mid x)$, leading to a reduction in diversity. To address this, we introduce an additional variable $z$, representing various “modes” of $z$. We propose explicitly modeling “mode selection” $p(z \mid c)$ before applying CFG in diffusion steps, ensuring that the mode distribution is not distorted by guidance (see Equation 8).
> - We propose using an AR model to learn mode selection $p(z \mid c)$, leveraging a synthetic condition-latent pair $(c, z)$ dataset. This can be viewed as **a form of knowledge distillation from pre-trained models**. However, we kindly ask the reviewer to consider the broader design and intent of our entire framework. Introducing the abstract latent variable $z$ strategically enhances the utilization of distilled knowledge. Without the latent modeling and sampling in our proposed framework, merely distilling from pre-trained models would not alone lead to the observed improvements in diversity. Our work provides a theoretical explanation from the CFG perspective on how explicitly modeling "mode selection" helps mitigate the issue of mode collapse under high CFG (Sec. 3.1).
>
> To address concerns about the reliance on distilled knowledge, we have included an alternative approach that constructs latent tokens without relying on additional knowledge.
> - Specifically, we train a model using **color clusters** as latent tokens. For each color channel (R, G, and B) within the range of 0-255, we equally segment it into eight clusters, resulting in a total of $8 \times 8 \times 8 = 512$ color clusters. Given an image, we resize it to 4x4 pixels and assign a color cluster ID to each pixel based on its RGB value. The image is then encoded into a sequence of color cluster IDs (e.g., "$C_1$#$C_2$#...#$C_{512}$"), with each $C_i$ representing a color cluster ID. This sequence serves as the condition for training our Kaleido diffusion.
> - In Fig.R.4 of our rebuttal PDF, we showcase images generated using color clusters as latent tokens on ImageNet. Our results demonstrate that, compared to the baseline MDM, our Kaleido diffusion can generate much more diverse images with latent tokens derived purely from color clustering. This highlights that Kaleido diffusion's capability to generate diverse images is **independent of distilled external knowledge**, confirming that our approach can produce varied images without the aid of any other pretrained models.
>
>
> [1] Adding Conditional Control to Text-to-Image Diffusion Models
>
> [2] Completing Visual Objects via Bridging Generation and Segmentation
>
> ----
>
> ## **W1: Quantitative Results / Q3: Quantitative measure of diversity**
> (**See general response**)
>
> ----
>
> ## **W4: Implementation Details**
> Thank you for your valuable feedback. To enhance the reproducibility of our work, in the revised version of the paper, we will provide more implementation details including the model parameters, training/inference hyperparameters, and any additional setup information necessary to replicate our results in the revised version.
>
> ----
> ## **Q1: Missing results on ImageNet?**
> The results in ImageNet can be found in Fig. 5, the top row of Fig. 7, and Fig. 10 and 11 in the Appendix. We will make this clear in the revised version that these figures illustrate the performance on the ImageNet.
>
> ----
> ## **Q2: The dataset used in Fig. 5**
> The dataset utilized for the results presented in Fig. 5 is ImageNet. We will ensure that this information is explicitly stated in the revised version

---

> ### Author Response · Authors · 2024-08-07
> **Dual submission policy**
>
> We would like to direct the reviewer’s attention to the dual submission policy outlined for NeurIPS [1]. According to this policy, “Papers previously presented at workshops are permitted, so long as they did not appear in a conference proceedings (e.g., CVPRW proceedings), a journal or a book.” Presentations at ICML workshops are not considered as part of the archival proceedings of the ICML conference.
>
> [1] https://neurips.cc/Conferences/2024/CallForPapers

---

> > ### Comment · Reviewer_6rHP · 2024-08-07
> > **Post-rebuttal response**
> >
> > Thank the authors for providing their rebuttal. Most of my concerns are addressed and I will increase my rating to 6.

---

> > > ### Author Response · Authors · 2024-08-12
> > > **Thank you!**
> > >
> > > Thank you and we are glad our rebuttal addressed your concerns. Let us know anything you still have in mind or unclear.
> > >
> > > Best regards,

---

### Official Review · Reviewer_d7gp · 2024-07-13

**Soundness:** 2
**Presentation:** 4
**Contribution:** 3
**Rating:** 6
**Confidence:** 4

**Summary:**

This paper improves the diversity of diffusion generation by incorporating autoregressive latent priors. It leverages the autoregressive model to generate specific discrete latent features, and then concat them with the original extracted text features to serve as the condition of diffusion model. Experiments show that the method can achieve high quality and diversity even with high CFG.

**Strengths:**

The investigated problem is interesting and critical. By incorporating the intermediate "mode" representation, the method can apply CFG after the "mode" is sampled, which alleviates the "mode collapse" phenomenon. The author introduces four specific modes which can also support fine-grained editing.

**Weaknesses:**

(1) The writing lacks training and inference details. During training, whether all the modes are used in each step or they are randomly selected? For the visual tokens, whether the latent features from SEED are directly used or the sequence IDs are re-embedded? During inference, the autoregressive model is responsible for generating the latent modes if I understand correctly. Then how different latent modes are generated in control especially the combination of them?

(2) Evaluation is poor. In the experiment section, only a figure is provided without providing numerical numbers. The main experiment setting is confused (class conditioned or text conditioned). The construction of the toy example is also not clearly stated.

(3) Comparison with related works is not sufficient, making the position of this work unclear. There are related works also trying to improve the generation diversity of diffusion models such as [a].

[a] CADS: Unleashing the diversity of diffusion models through condition-annealed sampling.

**Questions:**

I hope the authors can answer my questions in Weakness section point by point. I am willing to raise my score if my concerns are well addressed.

**Limitations:**

See Weakness.

---

> ### Author Rebuttal · Authors · 2024-08-07
>
> ## **W1: Training and Inference details**
>
> We appreciate your inquiry into the specifics of our training and inference methodologies.
>
> - During training, we investigate both isolated and combined uses of different latent tokens, including text, bounding boxes (bbox), blobs, and vokens, to highlight the unique contributions of each token type. Specifically, we train separate models focusing on individual latent types and a combined model that integrates text, bbox, and voken tokens for text-to-image generation.
>
> - Regarding the visual tokens, the latent tokens are formulated as a sequence of discrete image token IDs ("$I_1$#$I_2$#...#$I_{32}$"), where each $I_i$ denotes an image token ID (L754-756). Our autoregressive (AR) model re-embeds these image token IDs, resizing its vocabulary to accommodate the special image tokens (L203-204). The original SEED features are not used in our formulation.
>
> - In the combined setting, we include all latent tokens in a shared vocabulary and train them jointly. The AR model predicts **text | bbox | voken** sequentially, allowing later latents to be controlled by earlier ones. We observed that the "text" token first expands the semantic aspects (e.g., objects, behaviors) of the generation, the "bbox" specifically controls the spatial allocation of described objects, and the visual tokens control the global styles of the image. We plan to explore other combinations in future work.
>
> We will provide more detailed descriptions of the training and inference details in the revised manuscript to eliminate any ambiguity.
>
>
> ## **W2: Quantitative results and Toy examples**
>
> - The main experiment setting for the quantitative experiments is class-conditional generation on ImageNet 256x256, which aligns better with existing works. We also explored training text-to-image models with our methods with mainly qualitative comparisons.
>
> - In response to the concerns regarding quantitative evaluation, we expand our quantitative evaluation to include metrics like the Mean Similarity Score (MSS) and Vendi score, which are used for measuring image diversity following CADS. (**See general response**)
>
> - Regarding the toy example, we construct the toy dataset with two primary classes, each comprising two subclasses with a predefined weight (30% of samples in the first subclass and 70% in the second subclass). Each subclass is sampled from a Gaussian distribution. We train two models for comparison: a standard conditional diffusion model that uses the major class ID as conditions, and a latent-augmented conditional diffusion model that takes both the major class ID and subclass ID as conditions, with the subclass ID serving as latent priors. Both models are trained with classifier-free guidance. We design this toy experiment to show the benefit of latent priors for improving diversity under high guidance.
>
> We will include these elaborated details and the results from these models in the revised version of our paper.
>
>
>
> ## **W3: Comparison with related works**
>
> Thank you for suggesting a comparison of our work with relevant methodologies like CADS.
> - We have now included a comparison with CADS in our study for both class-to-image and text-to-image settings.
> - Our results demonstrate that both Kaleido and CADS can effectively enhance the generation diversity of diffusion models, particularly for class-to-image generation tasks like ImageNet, in a relatively orthogonal manner. Additionally, we show that the improvements from Kaleido and CADS can be complementary.
> - **Please refer to the general response to all reviewers for detailed results and discussion.**

---

> ### Author Response · Authors · 2024-08-12
> **Discussion Periods**
>
> Dear Reviewer,
>
> As the discussion period deadline nears, we would greatly appreciate it if you could review our rebuttal and share any further feedback. If there are still concerns, we would greatly appreciate a list of specific changes you would need to reconsider your rating.
>
> Thank you for your time and consideration.
>
> Best regards,

---

> ### Comment · Reviewer_d7gp · 2024-08-13
>
> Thanks for the clarification and new results. I would like to keep my positive rating.

---

> > ### Author Response · Authors · 2024-08-13
> >
> > Thank you for your valuable feedback! We hope our responses have clarified your concerns, and it would be nice if you consider raising your score to reflect this. However, thank you so much again!

---

### Official Review · Reviewer_7a2N · 2024-07-14

**Soundness:** 3
**Presentation:** 3
**Contribution:** 3
**Rating:** 7
**Confidence:** 5

**Summary:**

In this paper, the authors propose a principled pipeline called Kaleido Diffusion for text-to-image generation with better mode coverage and diversity. The main intuition is that, conventional DM requires large CFG to make samples locate at high-likelihood modes, which would somehow constrain the diversity by only generating those limited modes. To solve this problem, the authors propose to first use AR to capture the distribution of such latent tokens, then use diffusion model to take them as extra conditions for generation. Experiments on both quantitative numeraical results and qualitative visualizations are used to evaluate the effectiveness of this work.

**Strengths:**

1) The intuition of this work makes sense to me. The excessively large CFG would limit generated samples to certain modes, where using an extra model to capture such text-irrelevant information is a good practice.
2) The visual quality and especially the teaser of this work can well demonstrate the effectiveness of this paper.
3) The experiments are thorough to me.

**Weaknesses:**

1) I still don't well comprehend why choosing AR as the first-stage distribution learner. One of the reason I guess here is that, the CFG in AR model is only modulating the predicted logits, so that we can still sample across multiple reasonable modes for the next-stage DM through temperature parameter or top-k/top-p sampling. However, I'm still a little bit confused by this part: what if we use a diffusion model w/o CFG sampling to learn such an intermediate distribution? Is the model architecture design choice mainly for the fact that the authors use detailed textual descriptions as the latent token, so that it's natural to choose AR? If so, then this is a little bit ad-hoc to me.

2) Choosing these four specific types of latent tokens is also kind of ad-hoc to me. For example, somebody may also say the excessive CFG could result in similar artistic styles in generation. In this way, we have to additionally extract the style token as ground truth, then laboriously train the whole pipeline again to enable it. Another drawback of such design is that: the training cost is large and it's hard to scale up, every time we want to solve the mode collapse of new categories, we have to first extract/tokenize that new aspect, then jointly train the diffusion model and AR part, which is too ad-hoc and hard to scale up to me.

3) I hope the authors could also discuss about the diffusion decoder idea and the difference between that and this work. Specifically, AR + diffusion decoder would similarly take the latent tokens predicted by AR and feed into DM decoder as conditions. The major difference is that their setting doesn't need the ground truth for those latent tokens, which means the latent tokens are implicitly learned via end2end training. My question is what's the benefit of Kaleido compared to their pipeline, and won't their pipeline better at scaling up?

**Questions:**

They are mainly elaborated in the weakness section. Overall I really like the intuition of this work. I'd like to see the authors' responses on my questions.

**Limitations:**

The limitations are discussed in the appendix.

---

> ### Author Response · Authors · 2024-08-05
> **Request for clarification for W3 "the diffusion decoder idea"**
>
> Dear Reviewer,
>
> We are currently in the process of drafting our rebuttal response and would greatly appreciate your clarification on a point mentioned in Weakness 3. Specifically, we are seeking clarification on the "diffusion decoder idea" referenced in your feedback. At present, we believe you might be referring to approaches like multimodal large language models (MLLM) that employ diffusion models as decoders for image generation, similar to the approach used in EMU [1] [2].
>
> Could you please confirm if our understanding is correct? If not, we would be grateful if you could provide a reference to the specific paper or work related to the "diffusion decoder idea" so that we can address this point more accurately in our response.
>
> Thank you very much for your assistance!
>
> [1] EMU: Generative Pretraining in Multimodality
>
> [2] Generative Multimodal Models are In-Context Learners

---

> > ### Comment · Reviewer_7a2N · 2024-08-05
> >
> > Dear Authors:
> >
> > Thanks for letting me know about the un-clarified point. Yes, this is exactly what I mean about the "diffusion decoder idea", which would in general uses the output from first-stage model (e.g., AR in your setting) as input to a diffusion model for image/video generation. Hope this can make my concerns clear.
> >
> > Best

---

> > > ### Author Response · Authors · 2024-08-05
> > >
> > > Thank you for your prompt response! We will carefully compare our approaches with theirs in our rebuttal.

---

> ### Author Rebuttal · Authors · 2024-08-07
>
> ## **W1: Reason for choosing AR in modeling mode selection distribution**
>
> We appreciate your insights and queries regarding our choice of employing an AR model to model $p_\theta(z | c)$. Our choice is grounded in several pivotal considerations:
>
> - **Why discrete?**
> The modes $z$ that humans can perceive from an image are largely categorical, abstract, semantic, and high-level information. Such abstract semantics are more easily represented in **discrete** symbols. Moreover, modeling the modes as discrete latents further provides an explainable and editable mechanism for the image generation process. It allows the user to adjust the discrete latent codes before final image production, granting greater flexibility and control over the output. This capability is particularly beneficial in scenarios requiring detailed customization or iterative design processes.  We demonstrate the impact of sequential latent editing in Fig. 8.
> - **Why autoregressive?** Given our objective to model $z$ as abstract discrete tokens, an AR model emerges as the most suitable and convenient method for handling such discrete structures. The inherent design of AR models, which sample one token at a time conditioned on previous tokens, naturally supports the generation of diverse modes.
>
> Using a diffusion model w/o CFG to learn $p_\theta(z \mid c)$ could also serve as an alternative for learning intermediate distributions, however, it presents several challenges.
> - First, employing a diffusion model to model discrete abstract latents is a challenging and ongoing research area. Alternatively, representing these abstract semantics (i.e., modes) with continuous tokens raises fundamental questions about the characteristics and truth distribution of these abstract continuous latents. The challenge lies in accurately defining a distribution that authentically captures the complex, abstract semantics underlying the mode using continuous token.
> - Moreover, even if such a distribution could be defined, diffusion models typically demand high CFG to model it effectively, which circles back to our original challenge with high CFG. Nevertheless, exploring the potential of using diffusion models without CFG as the mode-selection learner remains an intriguing avenue for future research.
>
> Additionally, we would like to clarify that the AR model does not employ CFG during the sampling of latent tokens. The integration of $p_\theta(z \mid c)$ in Equation 8 works to push the updating direction towards the sampled modes $z$ at each step.
>
>
>
> ## **W2: Reason for choosing the four specific types of latent tokens**
> - We would like to clarify that we employ various types of latent tokens in order to explore the best ways for representing the modes, each type of latent tokens does not serve as a restricted representative of a particular "subset" of modes. It is important to clarify that our goal is **not** to exhaustively cover every conceivable category of diversity for image generation.  In our experiments, we choose text, bbox/blob, and vokens because they are useful for showing different controls.
> - Sometimes, a single representative type of latents like text is sufficient to generate samples that are diverse enough in terms of various aspects. For instance, Fig. 12 in the Appendix.
>
> Our proposed model is a general tool that can cover most of the aspects of diversity.
> - If users seek to create images that focus on the diversity of specific artistic styles, they can manually adjust the discrete latent tokens to reflect desired styles.
> - Alternatively, like other general text-to-image generation models, users may opt to employ techniques such as LoRA to fine-tune the model to achieve enhanced diversity within specific artistic styles.
>
> We will keep enhancing the clarity of these points in our revised paper until the methodological approach is clearly understood.
>
> ## **W3: Difference from AR + diffusion decoder; Scalability**
>
> In comparing our work with MLLMs using diffusion models as decoders, such as EMU [1][2] and MiniGPT5 [3],
> - A fundamental difference is that their "autoregressive" generation of latent tokens is based on regression, which is **deterministic** and typically produces similar images from the same input $c$. This deterministic nature in EMU and MiniGPT5 arises from their training objectives. EMU predicts visual representations $z’$ from the text input $c$, applying an image regression loss with encoded visual embeddings of image $x$ as the ground truth. MiniGPT5 uses an MSE loss to minimize the distance between generated image features $z’$ and the encoded caption feature of text input $c$.
>
> - In contrast, our Kaleido diffusion allows for greater variability by explicitly modeling $p_\theta(z \mid c)$ for diverse mode selection. This ability to generate diverse latents distinguishes our work, addressing challenges in generating varied, high-quality images under high CFG.
> Fig. R.5 in our rebuttal PDF shows that, unlike EMU, which produces nearly identical images for a given $c$, Kaleido generates varied samples from the same text condition, demonstrating superior diversity.
>
> - Unlike models that jointly train the encoder and diffusion model decoder, Kaleido uses pretrained discrete encoders, offering flexibility and efficiency by reducing training costs and complexity.
>
> Regarding scalability,
> - the total parameter count for our AR model and diffusion model is approximately 1.5B and 500M, respectively, which enables low computational cost.
> - Both AR and diffusion can be trained in parallel jointly, with ground-truth latents pre-extracted. The training cost is similar to the standard language model and diffusion model training.
>
>
> [1] EMU: Generative Pretraining in Multimodality
>
> [2] Generative Multimodal Models are In-Context Learners
>
> [3] MiniGPT-5: Interleaved Vision-and-Language Generation via Generative Vokens

---

> > ### Comment · Reviewer_7a2N · 2024-08-13
> >
> > I thank authors for the detailed reply. Overall I agree with the choice of AR to model the first-stage mode distribution. The reason for choosing those four specific types of latent tokens is mainly for paper demonstration. So this rebuttal has addressed most of my concerns. The only point is that I still think training separate tokenizer + accompanied AR model + diffusion makes the pipeline too complex to me. Even though I understand the model size is reasonable w/o need for too much compute, we still have to train the whole set of these three components every time we have a new requirement. But indeed this paper is clearly above the bar of this venue. I hence increase my score and advocate for acceptance of this work.

---

> > > ### Author Response · Authors · 2024-08-13
> > >
> > > Thank you for your valuable feedback and for increasing your ratings. We completely agree that further simplifying the pipeline to enhance scalability is crucial. This will be a key focus in our future work as well. Stay tuned!

---

> ### Author Response · Authors · 2024-08-12
> **Discussion Periods**
>
> Dear Reviewer,
>
> As the discussion period deadline nears, we would greatly appreciate it if you could review our rebuttal and share any further feedback.
> If there are still concerns, we would greatly appreciate a list of specific changes you would need to reconsider your rating.
>
> Thank you for your time and consideration.
> Best regards,

---

### Author Rebuttal · Authors · 2024-08-07

## **Quantitative Comparison with CADS:**

In response to the request for more quantitative results and comprehensive baseline comparisons, we have conducted additional experiments, specifically comparing our Kaleido diffusion model with CADS [1].

- **Condition Annealed Diffusion Sampler (CADS)** is a general sampling strategy that enhances the diversity of diffusion models by annealing the conditioning signal during inference.

- Following CADS, we employ two additional quantitative assessments of diversity: Mean Similarity Score (MSS) and Vendi scores. We use SSCD [2] as the pretrained feature extractor for calculating both MSS (SSCD) and Vendi (SSCD). Additionally, we utilize DiNOv2 [3] as the feature extractor for Vendi (DiNOv2), based on evidence from [4] suggesting that DiNOv2 provides a richer evaluation of generative models.

- Given that CADS is a training-free strategy applicable to different model architectures, we integrate CADS with both the baseline model MDM and our Kaleido-MDM. We emphasize that the contribution of CADS is **orthogonal** to our work, and its application is independent and complementary to the core methodologies in our research.
-----
## **Results**
We report the results for class-conditional generation on ImageNet 256×256 in Table 1 and 2, and for text-conditional generation on the MSCOCO [5] validation set in Table 3. All models use DDPM sampling with 250 steps.
- Table 1 presents a quantitative comparison based on evaluations from 50K samples. Our Kaleido diffusion outperforms the MDM + CADS combination in terms of FID-50K and precision, demonstrating that our method more effectively maintains high image quality while generating diverse samples. Furthermore, when we integrate CADS with our model, we achieve the best FID-50K results. Note that Precision cannot accurately evaluate models with diverse outputs since a model producing high-quality but non-diverse samples could artificially achieve high Precision [1].
- In Table 2 and 3, following CADS, we assess the diversity of the generated images using 10K samples. For Table 2, we select 1,000 random classes from ImageNet and generate 10 samples per class. For Table 3, we use 1,000 random text prompts from the MSCOCO validation set and generate 10 samples for each prompt. Our findings indicate that **both our Kaleido model and CADS significantly enhance sample diversity**. Although CADS achieves better performance in diversity, our model maintains superior image quality, as shown in Table 1.
- Additionally, **the methodologies used in CADS are complementary to ours, suggesting potential benefits from integrating CADS with our Kaleido model**. In fact, incorporating CADS into our model not only further improves image quality but also improves diversity, achieving the best scores in FID-50K, MSS (SSCD), and Vendi (DiNOv2) in class-conditioned image generation, and best Vendi (DiNOv2) in text-conditioned image generation.
- Lastly, our **rebuttal PDF** includes visual comparisons of these models for class- and text-conditioned image generation in Fig.R.1 and 2, respectively. All images are generated using DDPM with 250 steps. Specifically, in Fig.R.2, we observe that MDM + CADS fails to generate cats of diverse breeds from the prompt "a cat sleeping on the bed." In contrast, our Kaleido diffusion model excels, producing images of cats from various breeds with more diverse surrounding environments, showcasing its superior diversity capabilities. This observation contrasts with the trend of diversity scores in Table 2, suggesting that these diversity metrics may not fully capture certain aspects of diversity.
-----

**Table 1: Comparison on 50K samples of ImageNet, CFG=5.0**

| Model |                     FID-50K ↓ | Precision ↑ | Recall ↑ |
|----------|----------|----------|----------|
| MDM                              | 15.5       | **0.93**  | 0.22   |
| MDM + CADS                 | 10.6      | 0.60        | **0.62**   |
| Kaleido (ours)                  | 9.0       | 0.85        | 0.42  |
| Kaleido (ours) + CADS    | **5.9**  | 0.76       | 0.52 |

-----
**Table 2: Diversity Comparison on 1K x 10 Samples of ImageNet**

| Model               | MSS (SSCD) ↓ | Vendi (SSCD) ↑ | Vendi (DiNOv2) ↑ |
|------------------|----------|----------|----------|
| MDM                             | 0.21         | 8.42        |  3.04   |
| MDM + CADS                | **0.12**   | **9.28**  | 4.72   |
| Kaleido (ours)                 | 0.16        | 8.82        | 3.79   |
| Kaleido (ours) + CADS   | **0.12**   | 9.21       | **4.83**  |

-----
**Table 3: Diversity Comparison on 1K x 10 Samples of COCO Val**

| Model               | MSS (SSCD) ↓ | Vendi (SSCD) ↑ | Vendi (DiNOv2) ↑ |
|------------------|----------|----------|----------|
| MDM                             |    0.29           | 7.55            | 3.39  |
| MDM + CADS                |   **0.18**      | **8.65**      | 4.60   |
| Kaleido (ours)                 |  0.20           | 8.52            | 4.59   |
| Kaleido (ours) + CADS   |  0.19          | 8.61           | **4.75**  |

------
------

[1] CADS: Unleashing the diversity of diffusion models through condition-annealed sampling.

[2] A Self-Supervised Descriptor for Image Copy Detection.

[3] DINOv2: Learning Robust Visual Features without Supervision.

[4] Exposing flaws of generative model evaluation metrics and their unfair treatment of diffusion models.

[5] Microsoft COCO: Common Objects in Context

---

### Decision · Program_Chairs · 2024-09-25

**Decision:**

Accept (poster)

**Comment:**

This paper has received consistent feedback from four reviewers, all of whom are in favor of acceptance. The reviewers engaged in thorough discussion and rebuttal, ultimately reaching a consensus. This paper proposes Kaleido Diffusion for text-to-image generation. The method enhances the diversity of generated images, and the resulting latent variables are interpretable. Therefore,  the AC  has decided to accept this paper.